# Integrative common and rare variant analyses provide insights into the genetic architecture of liver cirrhosis

We report a multi-ancestry genome-wide association study on liver cirrhosis and its associated endophenotypes, alanine aminotransferase (ALT) and γ-glutamyl transferase. Using data from 12 cohorts, including 18,265 cases with cirrhosis, 1,782,047 controls, up to 1 million individuals with liver function tests and a validation cohort of 21,689 cases and 617,729 controls, we identify and validate 14 risk associations for cirrhosis. Many variants are located near genes involved in hepatic lipid metabolism. One of these, *PNPLA3* p.Ile148Met, interacts with alcohol intake, obesity and diabetes on the risk of cirrhosis and hepatocellular carcinoma (HCC). We develop a polygenic risk score that associates with the progression from cirrhosis to HCC. By focusing on prioritized genes from common variant analyses, we find that rare coding variants in *GPAM* associate with lower ALT, supporting *GPAM* as a potential target for therapeutic inhibition. In conclusion, this study provides insights into the genetic underpinnings of cirrhosis.

Cirrhosis of the liver results from prolonged hepatic inflammation and replacement of healthy liver tissue with scar tissue. It is an irreversible and progressive disease that is associated with high morbidity and mortality due to liver failure, cardiovascular and renal complications and a high rate of hepatic malignancies[1]. Major lifestyle and environmental risk factors for cirrhosis include chronic viral hepatitis, alcohol abuse and fatty liver disease. Mirroring the obesity pandemic, obesity-associated fatty liver disease is projected to soon become the most common cause of cirrhosis globally[1,2].

Estimates from twin studies have shown that approximately half of the variation in cirrhosis risk is attributed to genetic factors[3]. Identifying the implicated risk loci has progressed steadily over the last decade, mainly due to ever-larger genome-wide association studies (GWAS). The largest of these included 4,829 cases and 72,705 controls in the discovery cohort and identified 12 loci to be associated with cirrhosis in multitrait GWAS with plasma alanine aminotransferase (ALT), a biochemical marker of liver cell injury[4]. However, the number of sequence variants linked to cirrhosis is

low when compared to the hundreds of risk loci identified for other complex traits and diseases.

A better understanding of the genetic factors that predispose to cirrhosis may improve our ability to predict, prevent and ultimately treat the disease. Using polygenic risk scores (PRSs) that account for the influence of multiple risk loci may assist in identifying individuals who are at an increased risk of developing cirrhosis[5]. Furthermore, the discovery of genetic variants linked to a reduced risk of cirrhosis may provide potential molecular targets for pharmacological intervention[6–9].

The aims of this study were fourfold. First, we conducted meta-analyses of 12 cohorts comprising 18,265 cirrhosis cases and nearly 1.8 million controls, aiming to discover new risk loci for cirrhosis. Second, we conducted a range of analyses to elucidate the potential of PRSs to predict the onset and progression of cirrhosis. Third, we examined interactions of the genetic risk variants with alcohol consumption, adiposity and type 2 diabetes mellitus (T2D). Fourth, we used whole-exome sequencing data to gauge the expected effects of therapeutic inhibition of both known and new risk genes.

✉e-mail: jonasghouse@gmail.com; stefan.stender@regionh.dk

## Results

### Genome-wide association results

An overview of the study design is shown in Fig. 1. In stage 1, we performed a GWAS meta-analysis of nine studies, comprising 15,225 cases with cirrhosis and 1,564,786 controls of European ancestry (Supplementary Table 1). The genomic inflation factor ($\lambda_{GC}$) for the European-specific meta-analysis was 1.11 with linkage disequilibrium (LD) score regression (LDSC) intercept of 1.02 (s.e. = 0.007), indicating that the observed inflation is due to polygenicity. In stage 1, we identified 12 genome-wide significant variants (Fig. 1 and Supplementary Table 2), 5 of which have not been previously reported in a cirrhosis GWAS. In stage 2, we conducted a cross-ancestry fixed-effects meta-analysis with individuals of East Asian (9.9%), African American (1.2%), Hispanic (1.0%) and European (87.9%) ancestries (Supplementary Table 1), totaling 18,265 cases and 1,782,047 controls. In the cross-ancestry meta-analysis, we identified 15 variants, including 8 previously unreported variants (Fig. 1, Table 1 and Supplementary Table 3). Of the 15 unique variants identified in stage 2, 3 were specific to the cross-ancestry analysis, whereas 12 reached genome-wide significance in both stages (Supplementary Table 3). The *ALDH2* locus was driven by a missense variant (rs671, p.Glu504Lys), which is common in East Asian populations, but is rare or absent in other ancestries (Supplementary Table 3). Similarly, the missense variant in *SERPINA1* (rs28929474, p.Glu366Lys) was only present in Europeans. The following two variants showed heterogeneous effects across ancestries ($P < 0.003$): *PNPLA3* rs738408 and *TM6SF2* rs739846. We estimated the heritability (the proportion of variation that is attributed to common genetic variants) using LDSC. We found that the SNP-based heritability estimates were consistent between Europeans ($h^2 = 5.1\%$, 95% confidence interval (CI): 3.5–6.8) and East Asians ($h^2 = 2.7\%$, 95% CI: −2.7 to 8.1; Supplementary Table 4). We could not estimate heritability in African American and Hispanic samples because of limited statistical power (Supplementary Table 4). We also reappraised variants that have been previously linked to cirrhosis in GWAS or candidate gene studies but were not detected in stage 1 or 2 of our study (Supplementary Table 5). Of the nine variants, we found evidence to support association ($P < 0.05$) with cirrhosis for four (in *CENPW*, *TOR1B*, *MBOAT7* and *MAFB*).

### Endophenotype-driven analyses

We used liver enzyme GWAS associations as priors to enhance genomic discovery for liver cirrhosis. Analysis of up to 1 million individuals of European ancestry yielded 314 independent genome-wide signals for ALT, including 114 previously unreported (Fig. 1a and Supplementary Table 6), and 419 independent genome-wide signals for γ-glutamyl transferase (GGT), with 106 previously unreported (Fig. 1a and Supplementary Table 7). Of the 307 ALT and 403 GGT lead variants that were available in the cirrhosis datasets, 21 ALT and 20 GGT variants were associated with cirrhosis (false discovery rate (FDR) < 0.05; Table 1 and Supplementary Table 8). Nine variants were identified through both ALT- and GGT-informed analyses. Of 32 unique variants, 11 were genome-wide significant in the cirrhosis meta-analyses, 2 had been implicated in cirrhosis in prior GWAS (*TOR1B* rs7029757 and *MBOAT7* rs4806498)[4,10], whereas the remaining 19 have not been associated with cirrhosis before (Table 1 and Supplementary Table 8). Of the 21 ALT variants, 2 were directionally discordant with cirrhosis risk, specifically rs9663238 in *HKDC1* and rs79287178 in *TNFSF10* (Supplementary Table 8), whereas all 20 GGT variants had concordant direction of effects with cirrhosis. We found that a PRS using these 21 variants identified via endophenotype-driven analysis associated significantly with cirrhosis in the UK Biobank (UKB; odds ratio (OR): 1.15 per s.d., 95% CI: 1.11–1.20, $P = 1.8 \times 10^{-11}$) and Million Veteran Program (MVP, OR: 1.09 per s.d., 95% CI: 1.07–1.11, 1.20, $P = 3.6 \times 10^{-19}$), but contributed only little to the variance explained ($r^2_{UKB} = 0.2\%$ and $r^2_{MVP} = 0.1\%$; Supplementary Table 9).

### Validation

A total of 36 risk variants were identified through cirrhosis GWAS and/or the endophenotype-informed analysis, of which 35 were available for replication in the MVP cohort (21,689 cases and 617,729 controls). Replication of *ALDH2* rs671 was not possible due to its absence in non-East Asian populations. Of the 35 variants, 14 (40%) reached Bonferroni significance ($P < 1.4 \times 10^{-3}$ (0.05/35 variants)) and 6 were nominally significant ($P < 0.05$) with concordant directions of effect in MVP (Supplementary Table 10). Of the 20 variants associated with a $P < 0.05$ in MVP, 10 were initially identified in stage 1 or 2 of the GWAS, whereas 10 were identified solely via endophenotype-informed analyses. For the 15 variants that did not replicate at $P < 0.05$, we found a high level of concordance in the magnitudes and directions of effects between the two datasets (Pearson's $r^2 = 0.73$, $P = 1.8 \times 10^{-3}$; Supplementary Table 10).

### Phenome-wide association study (PheWAS)

To identify the mechanism by which a variant or gene is linked to a disease, it is important to comprehend the range of phenotypic consequences resulting from carrying a specific sequence variant. Here we tested each of the 36 risk variants identified in GWAS and/or endophenotype-driven analyses for association with 41 binary and quantitative traits. As expected, many of the variants were associated with risk factors for liver disease (for example, alcohol dependence and lipids) and/or with fatty liver disease (Fig. 2). For example, more than half of the variants ($n = 21/36$) were associated with non-high density lipoprotein cholesterol (non-HDL-C; FDR < 0.05), 16/36 variants were associated with liver fat and 18/36 variants were associated with a registry-based diagnosis of nonalcoholic fatty liver disease (NAFLD; Fig. 2 and Supplementary Table 11). Variants associated with either liver fat or NAFLD had directionally concordant effects on cirrhosis. Three variants (rs739846 in *TM6SF2*, rs80215559 in *HFE* and rs738408 in *PNPLA3*) associated with higher liver fat, but lower levels of non-HDL-C (Supplementary Table 11). Other variants were associated with proteins and metabolites produced by the liver, such as uric acid ($n = 23/36$), sex hormone-binding globulin ($n = 20/36$) and albumin ($n = 19/36$). Of the newly identified variants, rs1229984 in *ADH1B* associated with a lower risk of alcohol dependence, alcoholic liver disease (ALD), cardiovascular risk factors (for example, hypertension, body mass index (BMI)) and cardiovascular disease (for example, coronary artery disease). Similarly, rs1937455 in *PDE4B* was associated with a lower risk of alcohol dependence, lower GGT levels and lower BMI. rs9663238 in *HKDC1* associated with lower HbA1c levels and T2D risk. Other previously unreported variants associated with lipid traits and fatty liver disease, including rs2792735 in *GPAM*, rs2980888 in *TRIB1*, rs13389219 in *COBLL1*, rs8178824 in *APOH* and rs339969 in *ICE2* (Fig. 2 and Supplementary Table 11).

### Comparison of genetic effects on NAFLD, ALD and cirrhosis

Next, we compared the effect sizes on cirrhosis and NAFLD ($n_{cases} = 22,944$) of 18 previously reported NAFLD variants along with the 36 cirrhosis variants identified here (totaling 38 distinct variants). Eight variants had significantly higher effects on cirrhosis compared with NAFLD ($P$ value for heterogeneity ($P_{Het}$) < 0.05/38; Fig. 3a and Supplementary Table 12). Of those, we found that rs72613567 in *HSD17B13* and known risk variants in *SERPINA1* (p.Glu366Lys) and *HFE* (p.Cys282Tyr) exhibited stronger effects on cirrhosis than on NAFLD. Moreover, we found that the variants near *HKDC1, HLA-DQB1* and *MAMSTR* likely influence cirrhosis via pathways distinct from those related to fatty liver disease. We also found that variants in *TRIB1, TM6SF2* and *APOE* had stronger effects on NAFLD compared with cirrhosis, indicating that they may primarily exert their effect on cirrhosis via fatty liver disease. Variation in *GCKR* was strongly associated with NAFLD but had no effect on cirrhosis. The previously reported NAFLD variants p.Thr165Ala in *MTARC1* and p.Ile148Met in *PNPLA3* had proportional effects on cirrhosis. We then compared the effects of the 36 cirrhosis variants with their respective effects on ALD ($n_{cases} = 2,931$) and NAFLD ($n_{cases} = 22,944$; Fig. 3b). We found

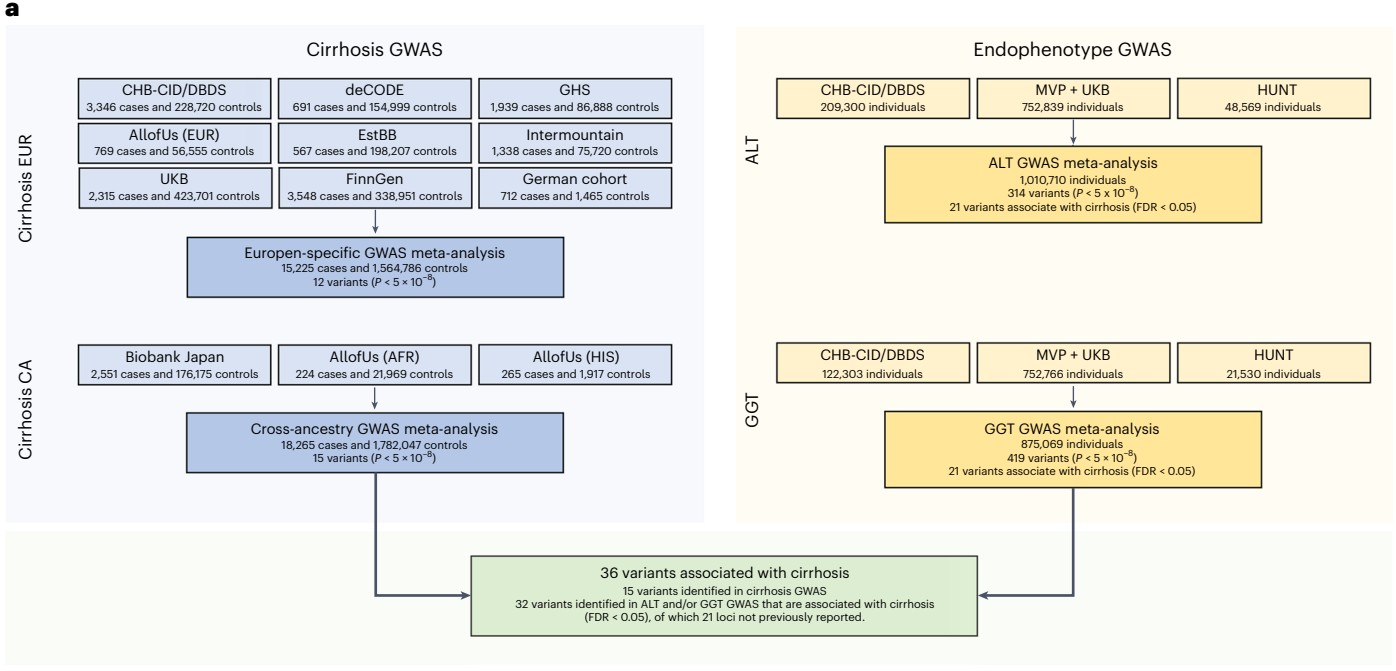

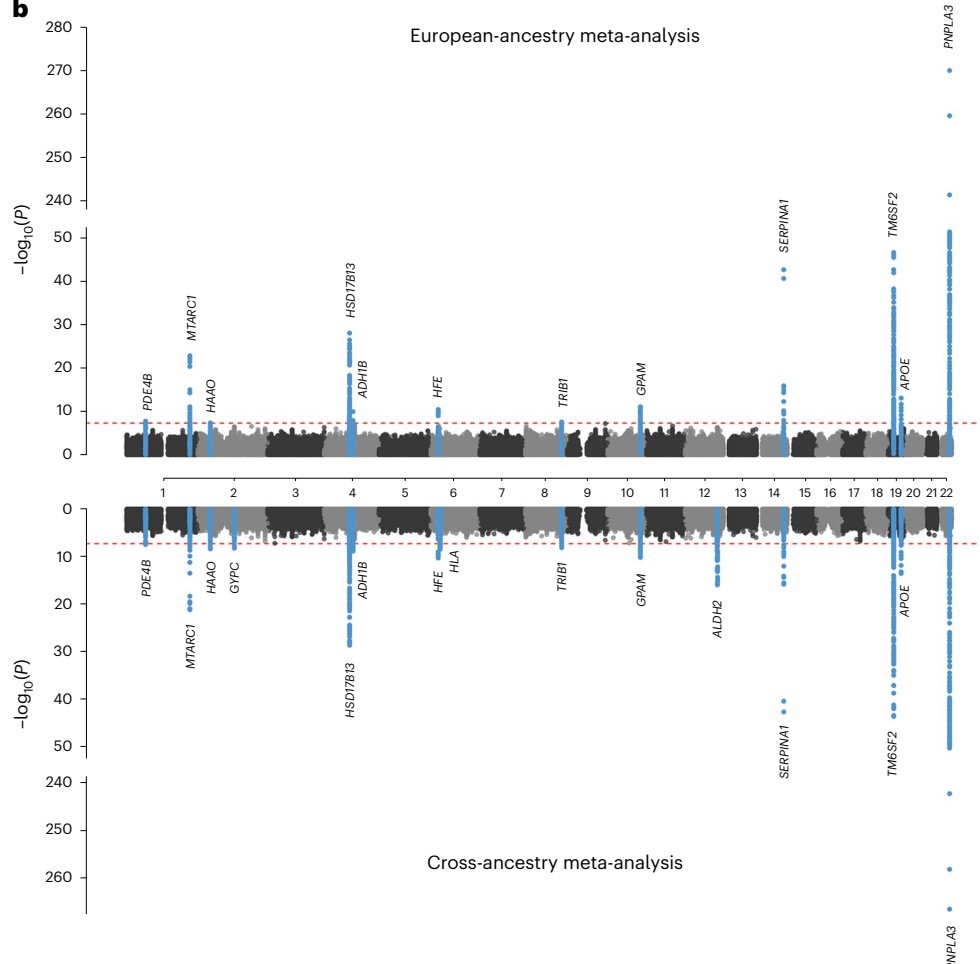

**Fig. 1 | Study design and main results. a,** Overall study design, with stage 1 representing the European-specific GWAS meta-analysis and stage 2 representing the cross-ancestry GWAS meta-analysis. **b,** Miami plot of cirrhosis GWAS. The *x* axis is the chromosomal location of SNPs and the *y* axis is the strength of association $-\log_{10}(P)$. Note that the *y* axis includes a break at 50. The lead SNPs and SNPs located within ±1 Mb are highlighted, and the nearest genes are annotated. The top plot shows results from the European-specific GWAS meta-analysis, whereas the bottom plot displays results from the cross-ancestry GWAS meta-analysis. The dashed red line represents the threshold for genome-wide significance ($P < 5 \times 10^{-8}$). *P* values were two-sided and based on an IVW fixed-effects meta-analysis, and not adjusted for multiple testing.

**Table 1 | Cirrhosis risk loci identified in the cross-ancestry GWAS meta-analysis and endophenotype-informed analyses**

| Locus | SNP | Chr:bp | Consequence | Nearest gene[a] | EA/ NEA | EAF gnomAD NFE | Meta-analysis CA | | | Known locus |
|---|---|---|---|---|---|---|---|---|---|---|
| | | | | | | | β | s.e. | P | |
| Cross-ancestry GWAS meta-analysis | | | | | | | | | | |
| 1 | rs7534143 | 1:66470379 | intron_variant | PDE4B | T/G | 0.459 | −0.0642 | 0.0116 | 3.27×10⁻⁸ | New |
| 2 | rs2642439 | 1:220970499 | intron_variant | MTARC1 | A/G | 0.319 | −0.1199 | 0.0125 | 6.21×10⁻²² | Known |
| 3 | rs10174815 | 2:43241234 | regulatory_region_variant | ZFP36L2/HAAO | C/A | 0.292 | 0.0715 | 0.0121 | 3.94×10⁻⁹ | New |
| 4 | rs404910 | 2:126994530 | intergenic_variant | GYPC | G/C | 0.165 | 0.0826 | 0.0142 | 5.63×10⁻⁹ | New |
| 5 | rs28636836 | 4:88231865 | intron_variant | HSD17B13 | T/C | 0.251 | −0.1419 | 0.0126 | 2.11×10⁻²⁹ | Known |
| 6 | rs17526590 | 4:100269045 | intron_variant | ADH1B | A/G | 0.110 | 0.1100 | 0.0181 | 1.34×10⁻⁹ | New |
| 7 | rs80215559 | 6:25918225 | intron_variant | SLC17A2/HFE | C/T | 0.059 | 0.1602 | 0.0244 | 4.70×10⁻¹¹ | Known |
| 8 | rs146650659 | 6:220970499 | intergenic_variant | HLA-DQB1 | G/A | 0.052 | 0.1472 | 0.0249 | 3.14×10⁻⁹ | New |
| 9 | rs2980888 | 8:126507308 | intron_variant | TRIB1 | T/C | 0.290 | 0.0692 | 0.0119 | 6.98×10⁻⁹ | New |
| 10 | rs2792735 | 10:113921825 | intron_variant | GPAM | G/A | 0.288 | 0.0776 | 0.0119 | 8.32×10⁻¹¹ | New |
| 11 | rs671 | 12:112241766 | missense_variant | ALDH2 | G/A | N/A | −0.2797 | 0.0337 | 1.03×10⁻¹⁶ | New |
| 12 | rs28929474 | 14:94844947 | missense_variant | SERPINA1 | T/C | 0.018 | 0.5000 | 0.0362 | 2.15×10⁻⁴³ | Known |
| 13 | rs739846 | 19:19419071 | intron_variant | SUGP1/TM6SF2 | A/G | 0.070 | 0.2598 | 0.0186 | 2.37×10⁻⁴⁴ | Known |
| 14 | rs483082 | 19:45416178 | upstream_gene_variant | APOC1/APOE | T/G | 0.226 | −0.1008 | 0.0132 | 2.68×10⁻¹⁴ | Known |
| 15 | rs738408 | 22:44324730 | synonymous_variant | PNPLA3 | T/C | 0.224 | 0.4116 | 0.0118 | 2.99×10⁻²⁶⁷ | Known |
| Endophenotype-informed analysis | | | | | | | | | | |
| 16 | rs2110944 | 2:37090233 | intron_variant | STRN | T/C | 0.483 | −0.038 | 0.011 | 8.34×10⁻⁴ | New |
| 17 | rs77375846 | 2:103155075 | downstream_variant | SLC9A4 | C/T | 0.116 | 0.062 | 0.019 | 8.80×10⁻⁴ | New |
| 18 | rs10164853 | 2:158481992 | intron_variant | ACVR1C | G/A | 0.074 | 0.062 | 0.019 | 1.31×10⁻³ | New |
| 19 | rs13389219 | 2:165528876 | intron_variant | COBLL1 | T/C | 0.394 | −0.042 | 0.012 | 5.17×10⁻⁴ | New |
| 20 | rs12633863 | 3:149211512 | intron_variant | TM4SF4 | G/A | 0.466 | −0.036 | 0.011 | 8.94×10⁻⁴ | New |
| 21 | rs79287178 | 3:172294500 | intron_variant | TNFSF10 | A/G | 0.033 | 0.124 | 0.034 | 2.95×10⁻⁴ | New |
| 22 | rs28712821 | 4:39413780 | intron_variant | KLB | G/A | 0.397 | −0.048 | 0.011 | 2.05×10⁻⁵ | New |
| 23 | rs12500824 | 4:77416627 | intron_variant | SHROOM3 | A/G | 0.341 | 0.041 | 0.011 | 2.48×10⁻⁴ | New |
| 24 | rs28431971 | 4:100487315 | intron_variant | MTTP | A/G | 0.241 | −0.042 | 0.013 | 1.50×10⁻³ | New |
| 25 | rs7667391 | 4:146785400 | intron_variant | ZNF827 | T/A | 0.160 | 0.053 | 0.014 | 2.37×10⁻⁴ | New |
| 26 | rs35611012 | 7:128564825 | intergenic_variant | IRF5 | T/C | 0.324 | −0.040 | 0.012 | 1.16×10⁻³ | New |
| 27 | rs7029757 | 9:132566666 | intron_variant | TOR1B | A/G | 0.086 | −0.098 | 0.019 | 2.67×10⁻⁷ | Known |
| 28 | rs1658425 | 10:60331547 | intron_variant | BICC1 | G/C | 0.491 | 0.036 | 0.012 | 2.07×10⁻³ | New |
| 29 | rs9663238 | 10:70983629 | intron_variant | HKDC1 | A/G | 0.292 | 0.060 | 0.013 | 1.57×10⁻⁶ | New |
| 30 | rs10887777 | 10:89807366 | downstream_variant | KLLN | C/T | 0.240 | 0.037 | 0.013 | 3.27×10⁻³ | New |
| 31 | rs3184504 | 12:111884608 | missense_variant | SH2B3 | T/C | 0.467 | 0.062 | 0.012 | 5.44×10⁻⁷ | New |
| 32 | rs339969 | 15:60883281 | intron_variant | ICE2 | C/A | 0.358 | −0.036 | 0.012 | 2.14×10⁻³ | New |
| 33 | rs8178824 | 17:64224775 | intron_variant | APOH | T/C | 0.027 | 0.118 | 0.039 | 2.27×10⁻³ | New |
| 34 | rs11666792 | 19:49227043 | intron_variant | RASIP1/MAMSTR | A/G | 0.491 | 0.051 | 0.012 | 3.49×10⁻⁵ | New |
| 35 | rs4806498 | 19:54674742 | intron_variant | TMC4/MBOAT7 | T/C | 0.425 | 0.044 | 0.011 | 5.69×10⁻⁵ | Known |
| 36 | rs113469203 | 20:25343258 | intron_variant | ABHD12 | G/A | 0.446 | 0.061 | 0.016 | 1.54×10⁻⁴ | New |

[a]Nearest protein-coding gene. The shown P values are two-sided, based on an IVW fixed-effects meta-analysis, and not corrected for multiple testing. The P value threshold for statistical significance was set at 5×10⁻⁸ for the GWAS analysis. We used an FDR<0.05 to define statistical significance in the endophenotype-informed analysis (Supplementary Table 8). Chr:Bp, chromosome:position on the Genome Reference Consortium Human Build 37 (hg19); CA, cross-ancestry; EA, effect allele; EAF, effect allele frequency; NEA, noneffect allele; NFE, non-Finnish Europeans.

proportional effects between NAFLD and ALD, except for three variants (p.His48Arg in *ADH1B*, $P_{Het} = 4.4 \times 10^{-14}$; rs28636836 in *HSD17B13*, $P_{Het} = 1.8 \times 10^{-4}$; and rs28712821 in *KLB*, $P_{Het} = 7.9 \times 10^{-4}$), which had significantly larger effects on ALD than on NAFLD ($P_{Het} < 0.05/36$).

**Mendelian randomization (MR)**

To explore potential causal relationships between significant PheWAS findings and cirrhosis risk, we performed MR analyses. Consistent with observational data, we found evidence to support a causal role of higher BMI ($P_{IVW} = 3.1 \times 10^{-16}$) and higher alcohol intake ($P_{IVW} = 3.3 \times 10^{-7}$) with increased risk of cirrhosis (see Supplementary Table 13 for summary of results). The results were not driven by the effect of individual variants (see Supplementary Figs. 1 and 2 for effect and leave-one-out plots). To evaluate the potential mediating effect of NAFLD on the association between higher BMI, alcohol intake and cirrhosis, we conducted multivariable Mendelian randomization (MVMR) analyses while accounting

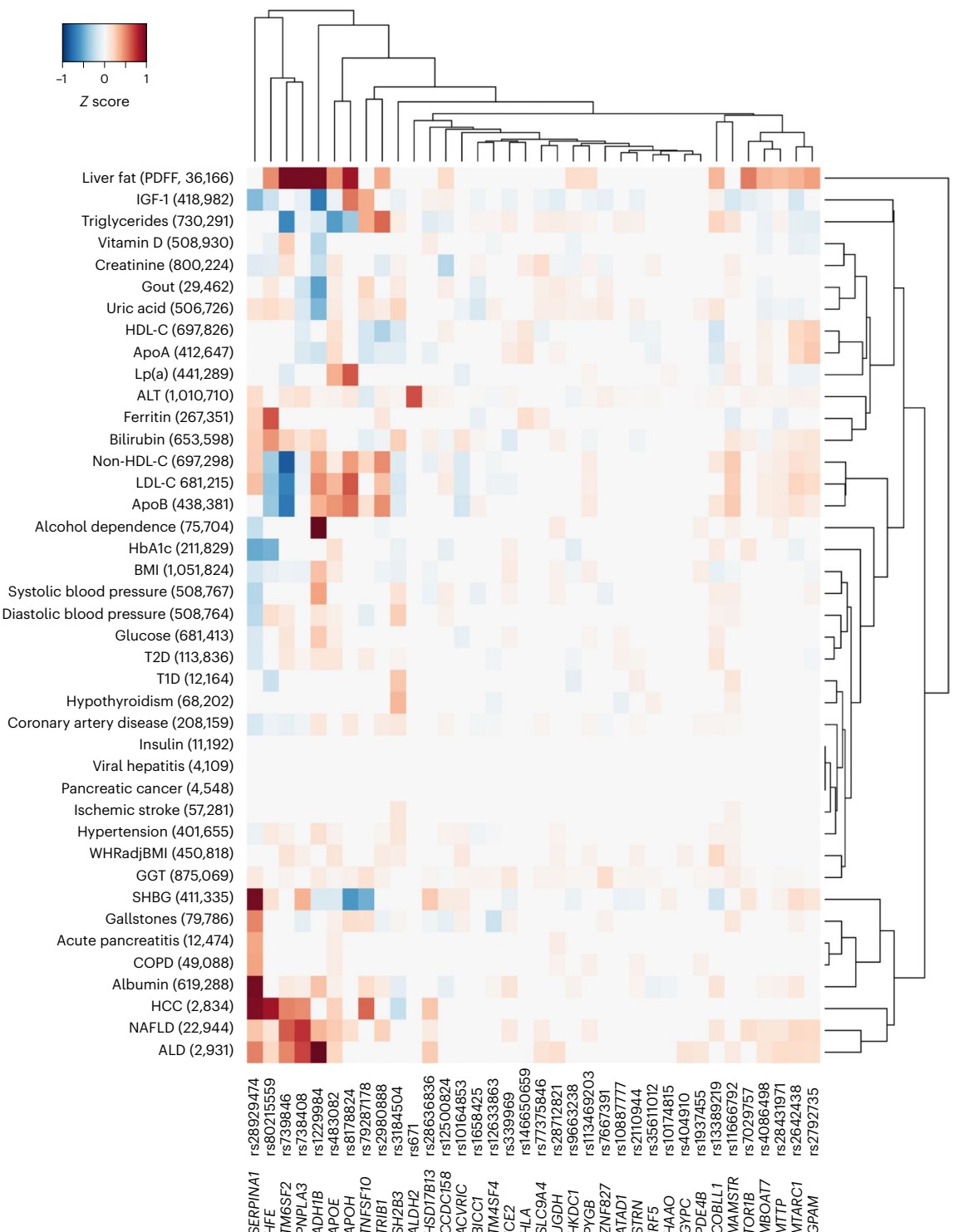

**Fig. 2 | Cross-trait associations between cirrhosis variants and selected metabolic and hepatobiliary traits relevant for cirrhosis.** The heatmap shows associations of variants identified through cirrhosis GWAS meta-analysis or endophenotype-driven analyses with 41 binary and quantitative traits sampled from meta-analysis of data from CHB-CID/DBDS, deCODE, Intermountain Healthcare, FinnGen, UKB and external sources, where available. The number of cases for binary traits and sample size for quantitative traits are shown in parenthesis following each trait. Shown are variants and phenotypes with significant associations after correcting for multiple testing using an FDR of <0.05. *P* values (two-sided) were derived from linear and logistic regression models. Hierarchical clustering was performed on a variant level using the complete linkage method based on Euclidian distance. Coloring represents *z* scores for each respective trait or disease, oriented toward the cirrhosis risk-increasing allele. Red indicates an increase in the trait or disease risk, while blue indicates a decrease in the trait or disease risk. SHBG, sex hormone-binding globulin; IGF-1, insulin growth factor 1; ApoA, apolipoprotein A; ApoB, apolipoprotein B; COPD, chronic obstructive pulmonary disease; WHRadjBMI, waist-to-hip-ratio adjusted for BMI; LDL-C, low-density lipoprotein cholesterol; T1D, type 1 diabetes.

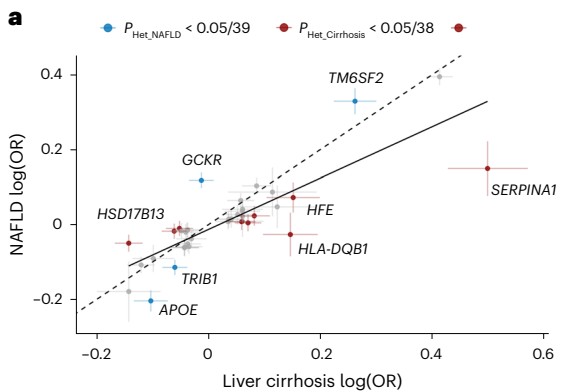
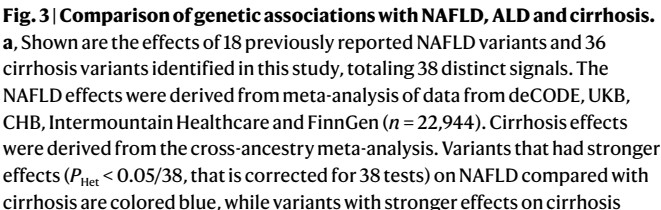
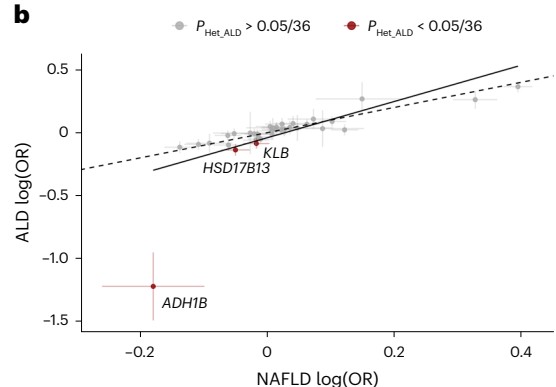

**Fig. 3 | Comparison of genetic associations with NAFLD, ALD and cirrhosis. a**, Shown are the effects of 18 previously reported NAFLD variants and 36 cirrhosis variants identified in this study, totaling 38 distinct signals. The NAFLD effects were derived from meta-analysis of data from deCODE, UKB, CHB, Intermountain Healthcare and FinnGen ($n = 22,944$). Cirrhosis effects were derived from the cross-ancestry meta-analysis. Variants that had stronger effects ($P_{Het} < 0.05/38$, that is corrected for 38 tests) on NAFLD compared with cirrhosis are colored blue, while variants with stronger effects on cirrhosis compared with NAFLD are colored red. **b**, Shown are the effects of the 36 cirrhosis variants on NAFLD ($n = 22,944$) and ALD ($n = 2,931$). Variants with stronger effects ($P_{Het} < 0.05/36$, that is corrected for 36 tests) on ALD compared with NAFLD are colored red. For both **a** and **b**, points refer to effect estimates (log(OR), measure of center), error bars represent 95% CI and the solid line represents the line of best fit. The dashed identity line ($y = x$) is shown for reference. $P_{Het}$ were two-sided and obtained using a likelihood ratio test (Cochran's $Q$).

for the influence of NAFLD. Despite observing a slight attenuation in the effect estimates, we observed significant independent associations between higher BMI ($\beta = 0.252$ s.d. units, s.e. = 0.048, $P_{IVW} = 2.7 \times 10^{-7}$) and alcohol intake ($\beta = 0.971$ s.d. units, s.e. = 0.186, $P_{IVW} = 1.3 \times 10^{-6}$) and risk of cirrhosis.

## Interactions with alcohol, obesity and T2D

Environmental risk factors and comorbid metabolic disorders, such as alcohol consumption, obesity and T2D exacerbate the impact of known genetic risk factors on cirrhosis[4,11]. To determine whether similar interactions exist between environmental factors and newly identified risk variants, we examined the effects of 35 genetic variants (excluding *HLA*) on cirrhosis risk in combination with environmental factors in the UKB. We found that rs738408 in *PNPLA3* interacted significantly with T2D ($P = 7.9 \times 10^{-6}$), BMI ($P = 3.0 \times 10^{-6}$) and weekly alcohol intake ($P = 1.2 \times 10^{-5}$) on the risk of cirrhosis (Supplementary Table 14). *PNPLA3* rs738408 was only weakly associated with T2D (OR: 1.03, $P = 0.007$), BMI ($\beta = -0.04$ kg m$^{-2}$, $P = 0.001$) and not associated with weekly alcohol intake ($\beta = -0.04$ units per week, $P = 0.160$). We then examined the common missense variant p.Ile148Met in *PNPLA3* (rs738409, $r^2 = 1$ with rs738408) and its interaction with the same environmental risk factors on a broader range of liver-related outcomes. We found that high alcohol intake (>14 units per week), obesity (BMI > 30 kg m$^{-2}$) and T2D also amplified the effect of *PNPLA3* p.Ile148Met on hepatocellular carcinoma (HCC) and all-cause liver disease (Fig. 4 and Supplementary Table 15). For instance, among obese individuals, homozygous carriers of the G-allele had a sevenfold increased risk of HCC compared to noncarriers. Among nonobese individuals (BMI < 30 kg m$^{-2}$), the corresponding risk was only 2.6-fold higher ($P$ for interaction = 0.003; Fig. 4 and Supplementary Table 15).

## Gene prioritization

To prioritize potential causal genes at the identified risk loci, we used the following six approaches: (1) identification of coding variants, (2) estimation of effects on gene expression using expression quantitative trait locus (eQTL) data from two datasets (GTEx v.8 and deCODE[12,13]), (3) associations with quantified splicing using splicing quantitative trait locus (sQTL) data from whole blood (deCODE), (4) effects on plasma protein levels using protein quantitative trait locus (pQTL) data from deCODE[14] and UKB[15], (5) a similarity gene-based method (polygenic priority score (PoPS)) and (6) Open Targets Variant-to-Gene (V2G) algorithm. Of the 36 cirrhosis variants, we identified protein-altering

variants in LD ($r^2 > 0.8$) with the lead variant at 16 loci (Supplementary Table 16), including 3 splice variants in *HSD17B13* (rs72613567, c.812+2dupT), *MAMSTR* (rs11666792, c.219+3G>A) and *PYGB* (rs2261790, c.1518+6T>C). Using gene-expression data, we found significant colocalization (posterior probability (PPa) >0.70) at six loci (Supplementary Tables 17 and 18), proposing 12 potentially causal genes, and a single gene at two loci (*HSD17B13* and *TOR1B*). Only *MBOAT7* and *HKDC1* showed evidence of colocalization in liver tissue. Additionally, we found six variants that associated with splicing QTLs (Supplementary Table 19) and two variants (in *ADH1B* and *APOH*) with significant *cis* associations with protein levels (Supplementary Table 20). Using the similarity-based approach, PoPS, we identified at least one gene at 23 loci that had a score among the top 10% of the PoPS distribution (Supplementary Table 21). Using the Open Targets Genetics V2G score, all variants were successfully mapped to a nearby gene. By considering the number of lines of evidence supporting a given gene, we found that 18 of 36 loci had at least two lines of evidence and 9 loci had at least three lines of evidence (Supplementary Table 22).

## Convergence between common and rare variant associations

We examined exome sequencing data in the UKB to assess convergence in disease risk between common and rare protein-truncating variants. We selected 18 genes based on gene-prioritization analyses that had at least two lines of evidence and then evaluated rare variants (allele frequency <0.1%) that were predicted to cause loss-of-function (pLoF) and/or missense variants (with a Combined Annotation Dependent Depletion (CADD) score of at least 20) for their association with ALT and cirrhosis. We found three genes (*ADH1B, GPAM* and *TM6SF2*) that were significantly associated with ALT ($P < 6.9 \times 10^{-4}$; Supplementary Table 23). Notably, rare pLoF variants in *GPAM* were associated with lower ALT levels (−0.29 s.d. units per allele, 95% CI: −0.40 to −0.16, $P = 5.8 \times 10^{-6}$) and numerically lower odds of cirrhosis, although the latter association did not reach statistical significance (OR: 0.36, 95% CI: 0.05–2.42, $P = 0.296$; Supplementary Table 23). In contrast, the missense variant p.Ile42Val (rs2792751) in *GPAM* had a positive effect on both ALT levels (0.006 s.d. units per allele, 95% CI: 0.005–0.007, $P = 7.0 \times 10^{-45}$) and cirrhosis (OR: 1.09, 95% CI: 1.06–1.12, $P = 6.4 \times 10^{-11}$). Similar to the missense variant p.Glu167Lys (rs58542926), rare coding variants in *TM6SF2* were also associated with higher ALT levels (0.10 s.d. units per allele, 95% CI: 0.07–0.13, $P = 2.0 \times 10^{-10}$) and increased risk of cirrhosis (OR: 2.07, 95% CI: 1.43–3.00, $P = 1.0 \times 10^{-4}$). The underlying mechanism by which the common missense variant p.Ile148Met

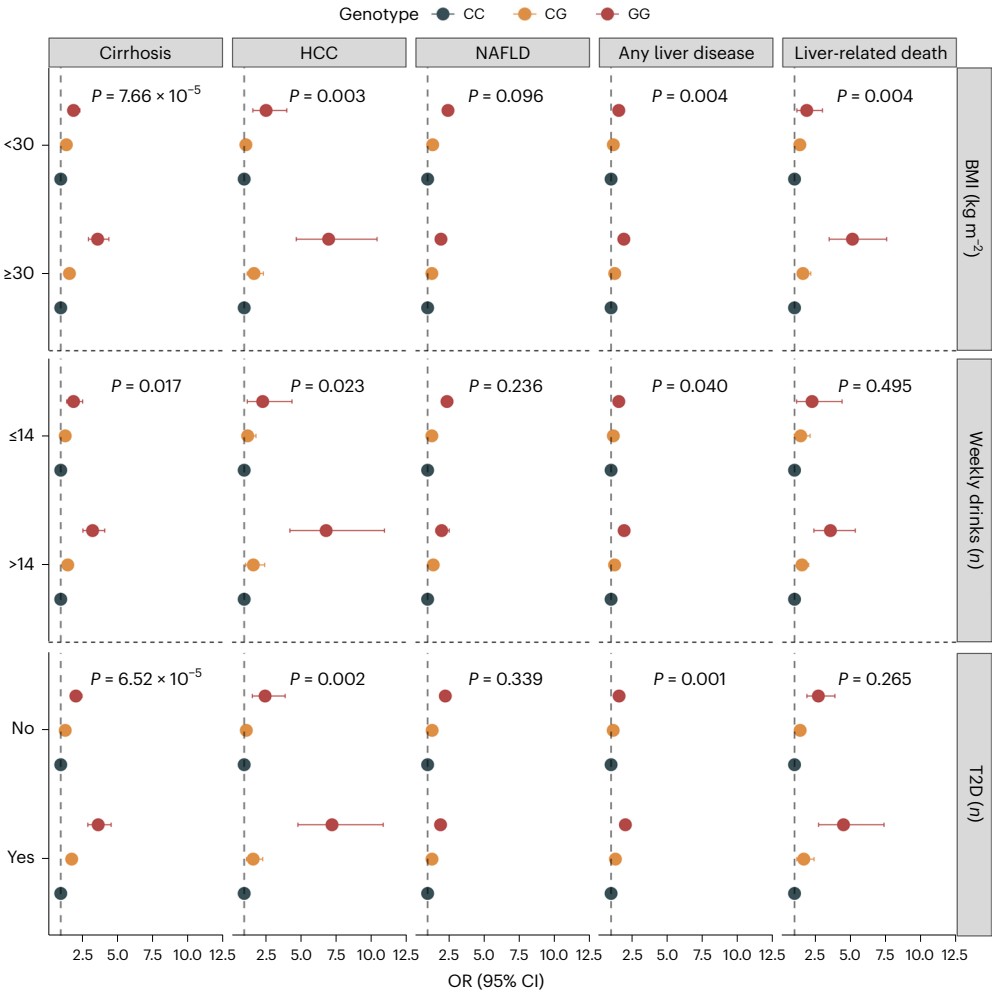

**Fig. 4 | Interaction between *PNPLA3* p.Ile148Met (rs738409), environment and risk of liver-related outcomes in the UKB.** Shown are the association between rs738409 carrier status and risk of five liver outcomes according to BMI categories (<30 versus ≥30 kg m⁻²), weekly alcohol intake (≤14 versus >14 weekly drinks) and T2D (no versus yes). Points refer to OR (measure of center), error bars represent 95% CIs and *P* represents *P* value for interaction. We used logistic regression models, adjusted for age, sex and ten principal components (PCs). Interactions between the variant and environmental factors were evaluated using likelihood ratio tests, comparing a main-effect model (variant + environmental factor) with a model including an interaction term (variant × environmental factor). The number of exposed individuals and the number of outcomes within each subcategory are listed in Supplementary Table 15.

(rs738409) in *PNPLA3* leads to hepatic steatosis and progressive liver injury has been a topic of discussion. We observed that rare coding variants (pLoF + missense) and pLoF variants (excluding missense) in *PNPLA3* were both nominally associated with increased cirrhosis risk (pLoF + missense−OR: 1.86; 95% CI: 1.19–2.90; $P = 6.0 \times 10^{-3}$; pLoF−OR: 2.97; 95% CI: 1.09–8.15; $P = 0.034$; Supplementary Table 23). We also found that rare coding variants in *PNPLA3* associated nominally with liver enzymes (pLoF + missense: 0.04 s.d. units per allele, 95% CI: 0.00–0.07, $P = 0.034$), but not when restricting to pLoF only (0.05 s.d. units per allele, 95% CI: −0.01 to 0.12, $P = 0.121$). This finding is similar to the direction of effect observed for p.Ile148Met (OR: 1.58, 95% CI: 1.54–1.62, $P = 3.1 \times 10^{-260}$). After adjusting for p.Ile148Met, associations were slightly attenuated (pLoF + missense− OR: 1.56; 95% CI: 1.04–2.33; $P = 0.032$; pLoF−OR: 2.01; 95% CI: 1.00–4.04; $P = 0.051$, respectively).

## PRS and hepatobiliary outcomes

We created the following six distinct PRSs: a European-specific (PRS$_{EUR}$), a cross-ancestry PRS (PRS$_{CA}$), a PRS based on ALT (PRS$_{ALT}$) and three different weighted scores, each incorporating varying numbers of risk variants identified in this study. We then compared the predictive ability of each of these PRSs. We found that the PRS$_{15-SNP}$ explained the highest

proportion of phenotypic variation ($r^2 = 1.7\%$; Supplementary Table 9), change in area under the curve (AUC) (+0.031, 95% CI: 0.023–0.039; Supplementary Table 24) and yielded an OR for cirrhosis of 1.42 per s.d. increase in PRS (Supplementary Table 9 and Fig. 5a). In comparison, the PRS$_{ALT}$ accounted for 1.3% of cirrhosis phenotypic variance, had a change in AUC of 0.021 and an OR of 1.38 per s.d. increase in PRS (Fig. 5a). The difference in predictive ability between PRS$_{15-SNP}$ and PRS$_{ALT}$ was statistically significant (change in AUC + 0.005, 95% CI: 0.003–0.017, $P = 0.005$). Next, we evaluated the reclassification of individuals after the addition of the PRS$_{15-SNP}$ to a baseline model containing age, sex and ten PCs. Adding PRS$_{15-SNP}$ resulted in a net percentage of individuals with cirrhosis correctly classified upward (event net reclassification index (NRI)) of 8.4% (95% CI: 3.1–13.7), and of individuals without cirrhosis correctly classified downward (nonevent NRI) of 21.3% (95% CI: 20.1–22.7). These changes resulted in an overall continuous NRI of 29.7% (95% CI: 23.4–36.1). Following this, we investigated how the various PRSs associated with a broader range of hepatobiliary outcomes. We found that the PRS$_{EUR}$ had the highest OR for HCC, for which a 1 s.d. higher PRS conferred an OR of 1.67 (95% CI: 1.52–1.82), followed by liver-related death (OR: 1.56 (95% CI: 1.44–1.69)) and alcoholic cirrhosis (OR: 1.47 (95% CI: 1.39–1.57)). Across the range of outcomes, the PRS$_{15-SNP}$ tended to have slightly larger per s.d. effect sizes than PRS$_{CA}$, PRS$_{ALT}$ and

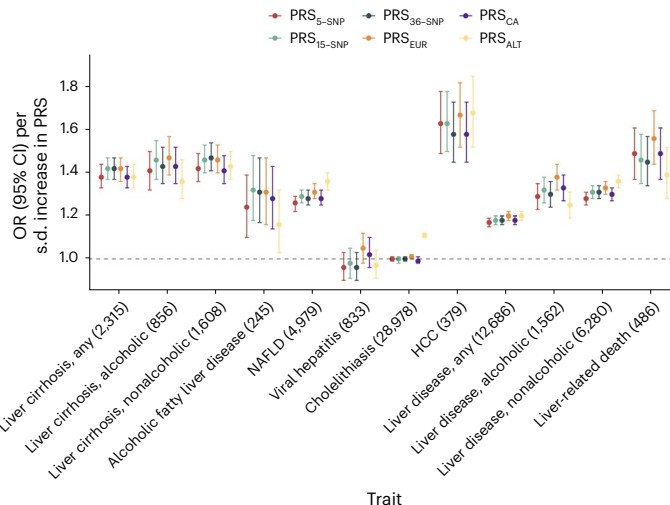

**Fig. 5 | PRSs and hepatobiliary outcomes.** Associations between six different PRS (PRS_ALT, PRS_5-SNP, PRS_15-SNP, PRS_36-SNP, PRS_EUR and PRS_CA) and hepatobiliary outcomes in the UKB. Points refer to ORs (measure of center) per s.d. increase in PRS; error bars represent 95% CIs. Total number of cases is provided following each outcome. Logistic regression models were used, adjusted for age, sex and ten PCs.

PRS_5-SNP, but comparable to the PRS_EUR. Notably, PRS_5-SNP performed similarly to PRS_ALT and PRS_CA, despite being based on only five SNPs. To test the generalizability of the PRS_5-SNP, we investigated its association with cirrhosis in a general population cohort from Copenhagen, Denmark (428 cases and 95,321 controls, all Danish ancestry), and in a multi-ancestry case–control study from Dallas, Texas (825 cases and 3119 controls; 21% Hispanic, 46% Black and 31% White). The per s.d. ORs for cirrhosis in the two cohorts were 1.35 (95% CI: 1.24–1.48) and 2.35 (95% CI: 2.10–2.63), respectively.

### PRSs and disease progression

We evaluated the ability of PRS_15-SNP to classify risk in a sample of 1,796 individuals with cirrhosis from the UKB, of whom 91 developed HCC. We found an association between a higher PRS_15-SNP and an increased risk of HCC after the onset of cirrhosis. Specifically, we found that individuals with cirrhosis and a high PRS_15-SNP (top 20% of the PRS) had a 10-year HCC risk of 15.0% (95% CI: 9.7–22.0) compared with 5.8% (95% CI: 4.3–7.6, P for difference <0.001) for individuals in the bottom 80% of the PRS_15-SNP (Fig. 6a). A similar pattern was observed in Copenhagen Hospital Biobank (CHB), involving 3,253 individuals with cirrhosis, of whom 172 developed HCC. Individuals in the top 20% of the PRS had an 11% (95% CI: 8.5–14.0) risk of developing HCC, compared to 5.3% (95% CI: 4.4–6.3, P for difference <0.001) for those in the bottom 80% (Fig. 6b). Correspondingly, the PRS associated with increased risk of progressing to cirrhosis in individuals with registry-defined NAFLD. We identified 4,449 individuals in the UKB with registry-defined NAFLD, of whom 193 progressed to cirrhosis during follow-up. Individuals with a PRS_15-SNP in the top 20% had a 10-year risk of 11.0% (95% CI: 7.1–16.0), whereas individuals in the bottom 80% of the distribution had a 10-year risk of 8.6% (95% CI: 6.8–11.0, P for difference = 0.036; Fig. 6c). In CHB, among 860 individuals with NAFLD, 95 developed cirrhosis during follow-up. In MVP, of the 18,302 individuals with NAFLD, 280 developed cirrhosis. Those in the top 20% of the PRS_15-SNP distribution had a 10-year cirrhosis risk of 13.0% (95% CI: 7.5–19.0) in CHB and a 5-year risk of 2.8% (95% CI: 2.3–3.5) in MVP, respectively (Fig. 6d). In contrast, those in the bottom 80% of PRS_15-SNP had a 10-year risk of 9.9% (95% CI: 7.6–12.0, P for difference = 0.032) in CHB and 5-year risk of 1.5% (95% CI: 1.3–1.7, P for difference <0.001) in MVP, respectively.

### Discussion

We report the largest GWAS meta-analysis to date on cirrhosis and its associated endophenotypes, ALT and GGT. Our study included over 18,000 cirrhosis cases and more than 1 million individuals with endophenotypic data sampled from four populations and identified 36 risk variants for cirrhosis, of which 14 replicated in an independent cohort. We found that PRSs were linked to the progression of NAFLD to cirrhosis and of cirrhosis to HCC. In addition, we used molecular QTLs and gene-prioritization methods to identify genes for rare variant burden analyses. This enabled us to investigate the convergence of risk between common and rare genetic variants and identify potential targets for pharmacological intervention.

As expected for an end-stage disease, we found that the risk variants were mainly associated with cirrhosis through known risk factors. The majority of the variants were associated with hepatic lipid metabolism and fatty liver disease, with certain variants (in *APOE* and *TRIB1*) displaying significantly larger effects on NAFLD compared with cirrhosis. Other variants, such as those in *HSD17B13* and *MAMSTR*, were found to have larger effects on cirrhosis compared with NAFLD, indicating a more dominant role in the progression to clinically advanced stages of chronic liver disease. Variants near *HKDC1* were mainly associated with cirrhosis, indicating the involvement of potential profibrotic pathways that do not involve the accumulation of hepatic fat. Conversely, other variants in *COBLL1* and *SH2B3* were related to body fat distribution traits, indicating that an impaired ability to store adipose tissue in peripheral compartments may contribute to disease[16–18]. Additionally, lead variants at *ADH1B* and *ALDH2* have been shown to cause adverse symptoms with alcohol intake, thus reducing the risk of alcohol-related diseases such as cirrhosis[19]. The variants at *HFE* and *SERPINA1* cause hemochromatosis and α-1 antitrypsin deficiency, respectively, well-known risk factors for cirrhosis.

The discovery of naturally occurring loss-of-function variants associated with protection against liver disease has led to the identification of new therapeutic targets[6,9]. We showed that rare loss-of-function variants in *GPAM* associate with lower plasma ALT levels. This finding aligns with two recent reports on the relationship between rare loss-of-function variants and ALT levels[9,20]. *GPAM* encodes the mitochondrial isoform of glycerol-3-phosphate acyltransferase, an enzyme that catalyzes the first step of glycerolipid synthesis in the liver and adipose tissue. Genetic variation in *GPAM* has previously been associated with fatty liver disease through GWAS[20,21] and with cirrhosis, albeit not at genome-wide significance[21,22]. These observations support the inhibition of *GPAM* as a potential treatment for cirrhosis and related liver diseases like steatosis and steatohepatitis[20].

The potential use of PRSs for prognostication in individuals at risk of cirrhosis is a topic of major clinical interest. We evaluated the predictive ability of a range of differently constructed cirrhosis PRSs. The main finding of these analyses was that a PRS based on the 15 SNPs that associated with cirrhosis at GWAS significance in our study performed as well as scores based on all 36 SNPs identified in our study, or scores based on more than 1,000,000 SNPs. Of note, a PRS based on the five SNPs with the strongest associations with cirrhosis had only slightly attenuated effects compared to the abovementioned PRSs. Taken together, our PRS analyses indicate that the genetic architecture of cirrhosis is dominated by few, large-effect variants. Such an oligogenic model is consistent with recent GWAS findings, which showed that a 9-variant PRS for chronic ALT elevation, a proxy for NAFLD, had effects similar to those observed for a 77-variant PRS[23]. Interestingly, the cross-ancestry PRS, which was derived from the largest set of cirrhosis cases, performed worse than the European-specific genome-wide PRS in predicting cirrhosis. This contrasts with recent studies on other complex traits that have shown that incorporating a broader set of ancestries can improve prediction, even for European populations[24–27]. The difference in performance may reflect the diverse underlying causes of cirrhosis, where viral hepatitis is the leading cause

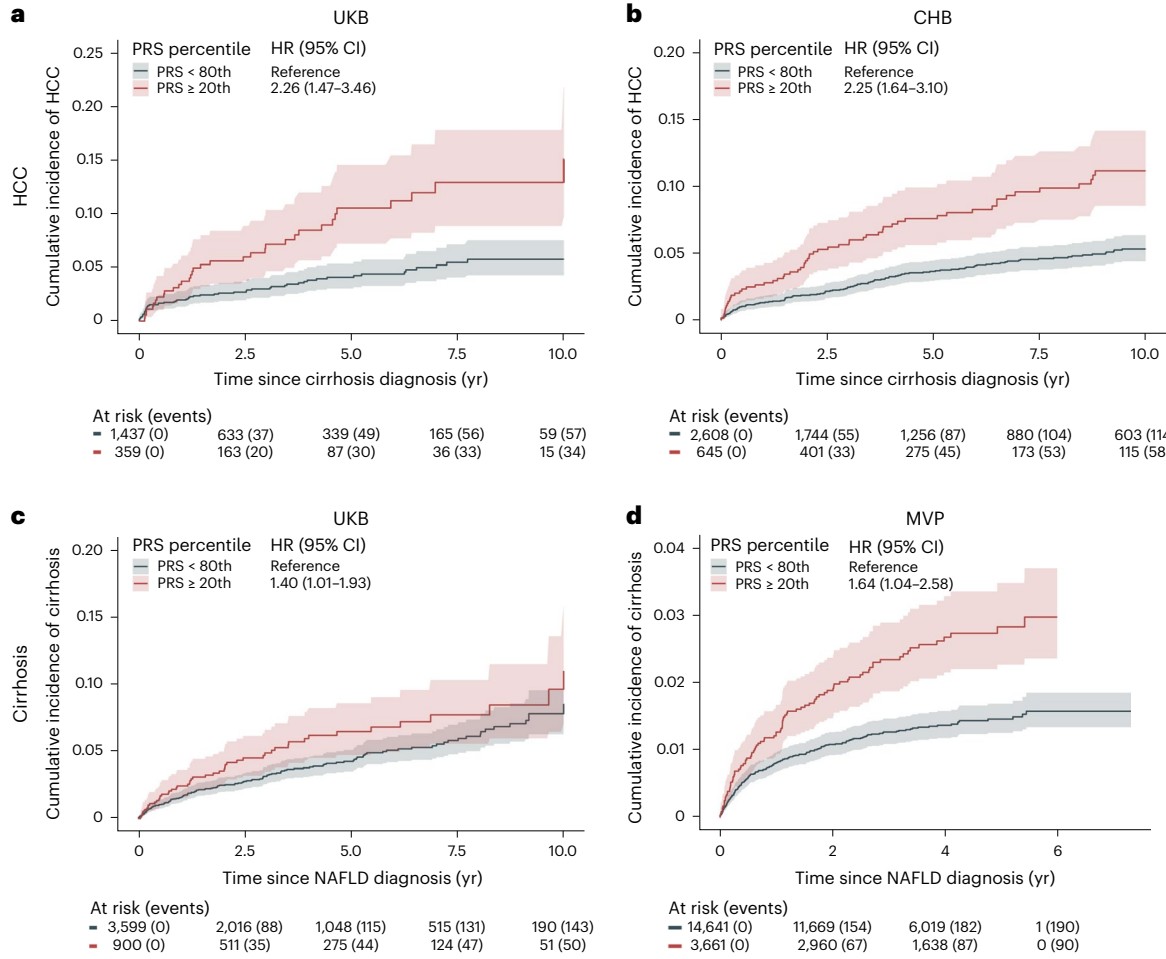

**Fig. 6 | PRSs and disease progression. a,b,** Risk of HCC in individuals with liver cirrhosis according to polygenic risk percentile. **c,d,** Risk of cirrhosis in individuals with NAFLD diagnosis, according to polygenic risk percentile. We used the PRS$_{15\text{-SNP}}$ to assign individuals to different risk groups. Cumulative incidence (solid line, measure of center) was estimated using Fine-Gray regression, which takes the competing risk of death into account. The lighter shades represent the respective 95% CIs. Number of individuals at risk according to each exposure group and events are given below each plot.

in East Asians, while alcohol and obesity are dominant in Europeans[2]. These differences are also reflected in the variants that are mainly driven by specific ancestries, such as the *HLA* locus association, which contributes to hepatitis persistence and chronicity in East Asians[28].

Environmental risk factors, such as BMI, exacerbate the risk of liver disease conferred by known genetic risk factors[11]. In alignment with previous observations[4,11], we found that the risk conferred by *PNPLA3* rs738409 was significantly amplified by adiposity, alcohol intake and diabetes for a range of liver-related outcomes, including NAFLD, cirrhosis and HCC. These relationships are among the strongest gene–environment interactions seen in man[29,30].

Our study has some limitations that should be considered. First, the cirrhosis phenotype was mainly based on registry-based International Classification of Diseases (ICD) codes, a definition that inevitably suffers from some degree of misclassification. That said, cirrhosis is a hard endpoint with well-defined diagnostic criteria. Supporting the validity of the endpoint, *PNPLA3* rs738409 was associated with cirrhosis in each cohort, with effect sizes like those seen in histologically defined cirrhosis cohorts. Second, we included relatively few individuals of African American and Hispanic ancestry. Third, although we included liver tissue in our eQTL analyses, the majority of eQTLs that we report are based on datasets from nonhepatic tissues, some of which had manyfold larger sample sizes. This limits the ability to draw conclusions on liver expression specificity of the reported loci. Nevertheless, we did not solely depend on eQTL signals as standalone evidence but

reported potential effector genes when complementary evidence from other gene mapping strategies converged on the same gene.

In conclusion, we identified 36 risk variants for cirrhosis, including 24 that have not been previously linked to this disease. These results provide an expanded catalog of genes to interrogate mechanistically in future studies. A better understanding of the genetic factors that underpin cirrhosis will improve our ability to predict and ultimately treat this deadly disease.

## Online content

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

Jonas Ghouse ⓘ[1,2] ✉, Gardar Sveinbjörnsson ⓘ[3], Marijana Vujkovic ⓘ[4,5,6], Anne-Sofie Seidelin[7], Helene Gellert-Kristensen ⓘ[7], Gustav Ahlberg ⓘ[2], Vinicius Tragante ⓘ[3], Søren A. Rand ⓘ[1,2], Joseph Brancale ⓘ[8], Silvia Vilarinho ⓘ[8], Pia Rengtved Lundegaard ⓘ[2], Erik Sørensen[9], Christian Erikstrup ⓘ[10], Mie Topholm Bruun ⓘ[11], Bitten Aagaard Jensen[12], Søren Brunak ⓘ[13], Karina Banasik ⓘ[14], Henrik Ullum[15], DBDS Genomic Consortium*, Niek Verweij[16], Luca Lotta ⓘ[16], Aris Baras[16], Regeneron Genetics Center*,**, Tooraj Mirshahi[17], David J. Carey[17], Geisinger-Regeneron DiscovEHR Collaboration*,**, VA Million Veteran Program*,**, David E. Kaplan ⓘ[4,5], Julie Lynch ⓘ[18,19], Timothy Morgan[20,21], Tae-Hwi Schwantes-An[20,22], Daniel R. Dochtermann[23], Saiju Pyarajan[23,24], Philip S. Tsao[25,26], Estonian Biobank Research Team*,**, Triin Laisk ⓘ[27], Reedik Mägi[27], Julia Kozlitina ⓘ[28], Anne Tybjærg-Hansen[7], David Jones[29], Kirk U. Knowlton[30,31], Lincoln Nadauld[29,32], Egil Ferkingstad ⓘ[3], Einar S. Björnsson[33,34], Magnus O. Ulfarsson[3,35], Árni Sturluson[3], Patrick Sulem ⓘ[3], Ole B. Pedersen ⓘ[36,37], Sisse R. Ostrowski ⓘ[9,36], Daniel F. Gudbjartsson ⓘ[3,38], Kari Stefansson ⓘ[3], Morten Salling Olesen ⓘ[1,2], Kyong-Mi Chang[4,5], Hilma Holm ⓘ[3], Henning Bundgaard[1,36,39] & Stefan Stender ⓘ[7,36,39] ✉

Article

[1]Department of Cardiology, Rigshospitalet, Copenhagen University Hospital, Copenhagen, Denmark. [2]Cardiac Genetics Group, Department of Biomedical Sciences, University of Copenhagen, Copenhagen, Denmark. [3]deCODE Genetics/Amgen, Reykjavik, Iceland. [4]Corporal Michael J. Crescenz VA Medical Center, Philadelphia, PA, USA. [5]Department of Medicine, University of Pennsylvania Perelman School of Medicine, Philadelphia, PA, USA. [6]Department of Biostatistics, Epidemiology and Informatics, Perelman School of Medicine, University of Pennsylvania, Philadelphia, PA, USA. [7]Department of Clinical Biochemistry, Rigshospitalet, Copenhagen University Hospital, Copenhagen, Denmark. [8]Section of Digestive Diseases, Department of Internal Medicine, and Department of Pathology, Yale School of Medicine, New Haven, CT, USA. [9]Department of Clinical Immunology, Rigshospitalet, Copenhagen University Hospital, Copenhagen, Denmark. [10]Department of Clinical Immunology, Aarhus University Hospital, Aarhus, Denmark. [11]Department of Clinical Immunology, Odense University Hospital, Odense, Denmark. [12]Department of Clinical Immunology, Aalborg University Hospital, Aalborg, Denmark. [13]Translational Disease Systems Biology, Novo Nordisk Foundation Center for Protein Research, Faculty of Health and Medical Sciences, University of Copenhagen, Copenhagen, Denmark. [14]Department of Obstetrics and Gynaecology, Copenhagen University Hospital Hvidovre, Copenhagen, Denmark. [15]Statens Serum Institut, Copenhagen, Denmark. [16]Regeneron Genetics Center, Regeneron Pharmaceuticals Inc, Tarrytown, NY, USA. [17]Department of Molecular and Functional Genomics, Geisinger Health System, Danville, PA, USA. [18]VA Informatics and Computing Infrastructure (VINCI), VA Salt Lake City Health Care System, Salt Lake City, UT, USA. [19]Division of Epidemiology, Department of Internal Medicine, University of Utah School of Medicine, Salt Lake City, UT, USA. [20]Gastroenterology Section, Veterans Affairs Long Beach Healthcare System, Long Beach, CA, USA. [21]Department of Medicine, University of California, Irvine, CA, USA. [22]Department of Medical and Molecular Genetics, Indiana University, Indianapolis, IN, USA. [23]Center for Data and Computational Sciences, VA Boston Healthcare System, Boston, MA, USA. [24]Department of Medicine, Brigham Women's Hospital and Harvard Medical School, Boston, MA, USA. [25]Palo Alto Epidemiology Research and Information Center for Genomics, VA Palo Alto, Palo Alto, CA, USA. [26]Department of Medicine, Stanford University School of Medicine, Stanford, CA, USA. [27]Estonian Genome Centre, Institute of Genomics, University of Tartu, Tartu, Estonia. [28]Eugene McDermott Center for Human Growth and Development, University of Texas Southwestern Medical Center, Dallas, TX, USA. [29]Precision Genomics, Intermountain Healthcare, Saint George, UT, USA. [30]Intermountain Medical Center, Intermountain Heart Institute, Salt Lake City, UT, USA. [31]University of Utah, School of Medicine, Salt Lake City, UT, USA. [32]Stanford University, School of Medicine, Stanford, CA, USA. [33]Faculty of Medicine, University of Iceland, Reykjavik, Iceland. [34]Internal Medicine and Emergency Services, Landspitali—The National University Hospital of Iceland, Reykjavik, Iceland. [35]Faculty of Electrical and Computer Engineering, University of Iceland, Reykjavik, Iceland. [36]Department of Clinical Medicine, University of Copenhagen, Copenhagen, Denmark. [37]Department of Clinical Immunology, Zealand University Hospital, Køge, Denmark. [38]School of Engineering and Natural Sciences, University of Iceland, Reykjavik, Iceland. [39]These authors jointly supervised this work: Henning Bundgaard, Stefan Stender. *Lists of authors and their affiliations appear at the end of the paper.✉e-mail: jonasghouse@gmail.com; stefan.stender@regionh.dk

## DBDS Genomic Consortium

Kari Stefansson[3], Erik Sørensen[9], Christian Erikstrup[10], Mie Topholm Bruun[11], Bitten Aagaard Jensen[12], Søren Brunak[13], Karina Banasik[14], Henrik Ullum[15], Ole B. Pedersen[36,37], Sisse R. Ostrowski[9,36] & Daniel F. Gudbjartsson[3,38]

Full lists of members and their affiliations appear in the Supplementary Information.

## Regeneron Genetics Center

Luca A. Lotta[16], Aris Baras[16] & Niek Verweij[16]

Full lists of members and their affiliations appear in the Supplementary Information.

## Geisinger-Regeneron DiscovEHR Collaboration

Tooraj Mirshahi[17] & David J. Carey[17]

Full lists of members and their affiliations appear in the Supplementary Information.

## VA Million Veteran Program

Marijana Vujkovic[4,5,6], David E. Kaplan[4,5], Julie Lynch[18,19], Timothy Morgan[20,21], Tae-Hwi Schwantes-An[20,22], Daniel R. Dochtermann[23], Saiju Pyarajan[23,24], Philip S. Tsao[25,26] & Kyong-Mi Chang[4,5]

Full lists of members and their affiliations appear in the Supplementary Information.

## Estonian Biobank Research Team

Triin Laisk[27] & Reedik Mägi[27]

Full lists of members and their affiliations appear in the Supplementary Information.

## Methods

### Ethics approval

All human research was approved within each contributing study by the relevant institutional review board (IRB) and conducted according to the Declaration of Helsinki (CHB-CID/DBDS: National Committee on Health Research Ethics; deCODE: National Bioethics Committee; Intermountain Healthcare: Intermountain Healthcare IRB; UKB: Northwest Multicenter Research Ethics Committee; Geisinger DiscovEHR: The GHS project has received ethical approval from the Geisinger Health System IRB under project 2006-0258; FinnGen: The Coordinating Ethics Committee of the Hospital District of Helsinki and Uusimaa; Estonian Biobank: ethical approval 1.1-12/624 from the Estonian Committee on Bioethics and Human Research, Estonian Ministry of Social Affairs; Biobank Japan: research ethics committees at the Institute of Medical Science, the University of Tokyo, the RIKEN Yokohama Institute and the 12 cooperating hospitals; Copenhagen General Population Study and Copenhagen City Heart Study: IRBs and Danish ethical committees; All of Us: National Institute of Health All of Us IRB; Dallas Liver Cohort: University of Texas Southwestern IRB; and MVP: VA Central IRB). All participants (except for CHB-CID) provided written informed consent. For CHB-CID, patients were informed about the opt-out possibility of having their biological specimens excluded from use in research in general. Since 2004, a national Register on Tissue Application (Vævsanvendelsesregistret) lists all individuals who have chosen to opt-out and whose samples cannot be used for research purposes. Before initiating this study, individuals listed in the Register on Tissue Application were excluded.

### Cohorts, association testing and meta-analysis

Cases were defined using hospital or registry records (ICD-9 or ICD-10). Controls were defined as individuals without a known history of cirrhosis. A full description of the cohorts and case and control definitions is provided in Supplementary Information and Supplementary Table 1. Details on genotyping methods, pre-imputation quality control and imputation methods are provided in Supplementary Table 1. Each study performed a GWAS of cirrhosis using logistic regression with at least age (or year of birth), sex and PCs used as covariates. Postregression quality control (QC) included the removal of variants with an imputation quality score <0.6, minor allele count <6 or absolute log(OR) or s.e. >10. We conducted two-fixed effect inverse-variance-weighted (IVW) meta-analyses using METAL[31]. The first involved individuals of European ancestry, including nine studies, totaling 15,225 cases and 1,564,786 controls. Only variants that were present in at least three studies were retained. In the second meta-analysis, we included individuals from East Asian (Biobank Japan), African American and Hispanic ancestries (latter two from All of Us), totaling 18,265 cases and 1,782,047 controls. Genomic inflation factors were calculated for each cohort and for the full meta-analysis. To assess any residual confounding due to population stratification, we calculated the LDSC intercept using LD scores calculated in the HapMap3 CEU population[32]. Genome-wide significance was set at $P < 5 \times 10^{-8}$.

### Risk loci definition

To identify independent variants within each risk locus, LD clumping was performed using PLINK v1.9. We used a 1 Mb window (--clump-kb 1000) and an LD threshold (--$r^2$ 0.1) to identify independently significant SNPs. Using the independently significant SNPs, distinct genomic loci were defined by starting with the lowest $P$ value variant, excluding other variants within ±1 Mb and iterating until no variants remained. The independently significant variant with the lowest $P$ value that defined each genomic locus is termed the lead variant. Risk loci were defined as a ±1 Mb region around each lead variant. A risk locus was termed new if neither the lead variant nor any variant within 1 Mb had previously reached genome-wide significance for cirrhosis.

### Endophenotype analyses

Both ALT and GGT levels are used clinically as biomarkers for liver injury. To increase statistical power for genomic discovery of cirrhosis, we used GWAS summary statistics for ALT and GGT as priors for association with cirrhosis. We first performed meta-analyses on both ALT and GGT summary data using previously published summary statistics[33,34] and data from CHB including more than 1 million individuals. We then tested independent variants that reached genome-wide significance for association with ALT or GGT in both the European-only and cross-ancestry cirrhosis meta-analysis. We considered associations significant if their FDR was <0.05.

### Validation

To validate our findings, we performed replication of cirrhosis variants identified via cirrhosis GWAS and/or endophenotype-informed analysis using summary statistics of 21,689 cirrhosis cases and 617,729 disease-free controls from the MVP. Two variants (rs146650659 and rs113469203) were not available in the MVP, for which we selected suitable proxies ($r^2 \geq 0.65$). rs671 in *ALDH2* was not amenable to validation, due to low frequency in non-East Asian populations. Cases were defined as in the primary GWAS analysis (Supplementary Table 1). A $P < 1.4 \times 10^{-3}$ (0.05/35 variants) and consistent direction of effect were considered successful replication.

### PheWAS

To gain insight into the potential underlying mechanisms by which the new risk loci contribute to disease, we tested the association between the 36 risk loci and 41 predefined metabolic and hepatobiliary traits using data from deCODE, UKB, FinnGen, Intermountain Healthcare, CHB-CID/DBDS and publicly available summary statistics, where available. The 36 variants were taken forward from the three main analyses (that is, the European-specific analysis, cross-ancestry meta-analysis and endophenotype-driven approach). In instances where risk loci were represented in multiple analyses, we selected the most significant variant (that is, the lowest $P$ value). Binary traits were analyzed using logistic regression, and quantitative traits were inverse-rank normalized and analyzed using linear regression. The models were adjusted for age, sex and ten PCs. We considered associations significant if their FDR was <0.05.

### Gene–environment interaction analyses

Environmental factors, such as alcohol consumption and BMI, are known risk factors for cirrhosis, and synergistic effects with genetic risk factors have previously been reported[4,11]. Here we systematically investigated for potential effect modification between risk loci and BMI, weekly alcohol intake and T2D in the UKB. BMI was measured at the baseline assessment visit and calculated as weight in kilograms divided by height in meters squared. Information on alcohol consumption was retrieved from questionnaire-based data on alcohol use. Participants who consumed alcohol at least once or twice per week were asked to provide information on their average weekly and monthly alcohol consumption across various alcoholic beverages (red wine, white wine, champagne, fortified wine, spirits and beer/cider). Based on data collected from individuals who consumed alcohol regularly, we calculated the average weekly alcohol intake in units. Information on T2D was retrieved from either from self-reported history of T2D or unspecified diabetes or HbA1c levels >48 mmol mol$^{-1}$ measured at baseline. We examined a total of 35 variants (excluding the *HLA* variant) from the three main analyses for interaction with these factors. Potential interactions between the variants and environmental factors were evaluated using likelihood ratio tests, comparing the main-effects model (variant + environmental factor) with a model including an interaction term (variant × environmental factor). We set the significance threshold at $P < 4.8 \times 10^{-4}$ (0.05/(35 variants × 3 traits)).

## MR

We conducted MR analyses on a set of biomarkers that had previously been identified as risk factors for cirrhosis, including BMI, lipids and alcohol intake[35]. To ensure that our analysis did not have overlapping samples, we conducted a meta-analysis on all available cirrhosis cohorts of European ancestry except for the UKB sample set, as all the exposure traits were derived from the UKB. We excluded exposure traits with fewer than ten instrumental variables (IVs) to avoid underpowered tests, resulting in 39 traits being tested. We evaluated instrument strength by calculating the $F$ statistic[36]. To ensure a comparable LD structure between exposure and outcome datasets, only exposures derived from samples of European ancestry were taken forward. We selected independent variants with genome-wide significance ($P < 5 \times 10^{-8}$) and an $r^2 < 0.001$ to serve as IVs for our MR analyses using the clumping procedure in the TwoSampleMR software and LD estimates from the European samples from the 1000 Genomes Project. We used the following two different MR methods: IVW model as our primary model and the weighted median model as sensitivity analysis. MR–Egger intercept was used to test for pleiotropy. To test whether the results were driven by individual variants, we conducted leave-one-out analyses. Only associations that passed $P < 1.2 \times 10^{-3}$ (0.05/39 traits) in the primary analyses (IVW), had a $P < 0.05$ in our sensitivity analyses (weighted median) and showed no evidence of pleiotropy (MR–Egger intercept $P \geq 0.05$) were considered significant. Finally, we explored whether the genetic effects of BMI and alcohol were mediated by the effect of NAFLD, by using the ivw_mvmr() function in the MVMR package. Genetic effects on NAFLD were obtained from a meta-analysis comprising 9,491 cases[20].

## Heritability

We used LDSC v.1.0.0 to estimate the SNP heritability of cirrhosis in Europeans, East Asians, African Americans and Hispanics using ancestry-matched precomputed LD scores obtained at https://gnomad.broadinstitute.org/downloads/. We reformatted association statistics to LDSC format with the munge tool, which excluded variants that did not match with the LD panel, had strand ambiguity, MAF < 0.01, INFO < 0.9 and variants that resided in long-range LD regions and the major histocompatibility locus on chromosome 6. To convert to liability scale, we used population-specific prevalence estimates, ranging from 0.5% in Europeans to 1.7% in East Asians[2].

## Gene mapping

We used six complementary approaches to annotate lead variants to potentially causal genes. First, we investigated whether the lead variants or proxy variants ($r^2 > 0.8$) were annotated as loss-of-function or missense variants using Variant Effect Predictor (VEP) v.95 (ref. 37). Second, we used molecular QTLs to investigate the relationship between risk loci and potential downstream effects on gene expression (eQTL), alternative splicing (sQTL) and protein levels (pQTL). We investigated whether lead or proxy variants overlapped with top cis-eQTLs from the following two resources: adipose ($n = 750$) and whole-blood eQTL ($n = 17,846$) data from deCODE[13] and 54 tissues and cell lines from GTEx (v.8)[12]. Top cis-eQTLs were eQTLs with the strongest association with each gene within a 1 Mb window and a $P < 1 \times 10^{-7}$. We used colocalization analyses to detect shared causal variants between cirrhosis and gene expression using COLOC (v.3.2.1) R package[38]. We tested genes with significant cis-eQTL association by analyzing all variants that were located within a ±1-Mb window around the sentinel variant using eQTL and cirrhosis, ALT and GGT meta-analysis summary statistics. We set the prior probabilities to $P_1 = 1 \times 10^{-4}$, $P_2 = 1 \times 10^{-4}$ and $P_{12} = 5 \times 10^{-6}$, as suggested previously[39]. We report the posterior probability that the association with gene expression and cirrhosis risk is driven by a single causal variant. We consider a PPa ≥ 0.70 as supporting evidence for a causal role for the gene as a mediator of cirrhosis. Data on alternative RNA splicing were derived from whole-blood RNA-seq ($n = 17,846$) data

available at deCODE[20]. The strongest association for each splice junction with a $P < 1 \times 10^{-8}$ was deemed top cis-sQTL. Data on protein levels were based on the following two datasets: (1) 4,907 proteins ($n = 35,559$) measured using the SomaScan v.4 assay available at deCODE[14] and (2) 1,472 proteins ($n = 47,151$) measured using the Olink Explore 1536 platform available at UKB[15]. Top cis-pQTLs were pQTLs that had the strongest association within a 1 Mb window. If the lead or proxy variants were in LD ($r^2 > 0.8$) with either a top cis-eQTL, top cis-pQTL or top cis-sQTL, the two signals were considered overlapping. Third, we used the gene that was assigned the highest Variant-to-Gene (V2G) score provided by Open Targets Genetics (https://genetics.opentargets.org/). The V2G score is an ensemble score that combines evidence on variant–gene associations from multiple sources, including molecular cis-QTL data (for example, pQTL and eQTL), interaction-based datasets (for example, promoter capture Hi-C) and genomic distance. For details on specific datasets and corresponding weights, please see https://genetics-docs.opentargets.org/our-approach/data-pipeline. Fourth, we used PoPS, a similarity-based gene-prioritization approach, which integrates GWAS summary statistics with gene-expression data, biological pathways and predicted protein–protein interaction data from more than 50,000 features[40]. We first computed gene-level association statistics and gene–gene correlations from our European-specific and cross-ancestry GWAS summary statistics using MAGMA[41] and LD estimates from 1000 Genomes European Ancestry data. Then, we conducted an enrichment analysis for gene features outlined at https://github.com/FinucaneLab/gene_features with MAGMA. Finally, we determined PoPS for each gene by fitting a joint model that considers the enrichment of all resulting features. Genes with a PoP score in the top 10% of the distribution were considered potential causal genes. We used the sum of the listed approaches and prioritized genes that had at least two lines of evidence. In the event of a tie-break, genes with coding variants in LD with the lead variant were given priority over V2G and/or PoPS.

## Convergence between common and rare variant associations

While common variant associations enable the connection between a specific gene region and a disease, associations with rare coding variants can precisely identify causal genes and offer insights into the potential therapeutic effect and direction of targeting a gene or its product. To investigate the directional concordance between common and rare variants, we conducted rare variant analyses, studying the association between rare pLoF and missense variants in genes, supported by our gene-prioritization analyses, and ALT and cirrhosis, respectively. Details on calling and quality control have been described elsewhere[42]. We used SnpEff to annotate the variants and prioritized those with a minor allele frequency of <0.1%, which were predicted to cause loss-of-function, including stop-gain, frameshift, splice acceptor and splice donor variants and missense variants with a CADD score ≥20. We created the following two masks: pLoF only and pLoF + missense with CADD ≥20. To evaluate the associations between genotypes and outcomes, we used linear regression models for quantitative traits (ALT) and Firth-bias corrected logistic regression models for binary traits (cirrhosis) using REGENIE and individuals of European ancestry[43]. The models were adjusted for age, sex and ten PCs. We set the significance threshold at $P < 6.9 \times 10^{-4}$ (0.05/(18 genes × 2 masks × 2 traits)).

## PRS derivation

The following six PRS were generated to compare the predictive performance in detecting cirrhosis: a cross-ancestry cirrhosis PRS ($n = 1,325,517$ (1.1% cases)), a European-only PRS ($n = 1,105,216$ (1.1% cases)), an ALT PRS ($n = 257,869$) and three weighted scores based on differing numbers of risk variants. Variants selected from the weighted scores were the 36 variants identified in both the cirrhosis GWAS and/or endophenotype-informed analysis, the 15 variants identified through cirrhosis GWAS and 5 known high-effect variants (rs2642438 in MTARC1,

rs72613567 in *HSD17B13*, rs28929474 in *SERPINA1*, rs739846 in *TM6SF2* and rs738408 in *PNPLA3*). Polygenic weights were calculated using PRS-CSx[44]. This method uses ancestry-specific GWAS weights, paired with LD information from an ancestry-matched external reference panel to estimate the posterior effect size for each SNP. Reference panels from the 1000 Genomes European, East Asian, Admixed American and African American samples were used. For the cross-ancestry PRS, ancestry-specific posterior effect sizes were meta-analyzed using the IVW method. For the genome-wide PRSs, we excluded the UKB dataset from the derivation datasets to ensure nonoverlapping samples. For the weighted scores, only effect estimates derived from meta-analysis excluding the UKB were used.

### PRS evaluation
We evaluated the PRSs in the UKB. We first evaluated the proportion of variance explained ($r^2$) by the six PRSs. We estimated the variance explained on the observed scale using Nagelkerke's $r^2$ as the difference in $r^2$ between a full model (PRS + sex + age + ten PCs) and a null model (sex + age + ten PCs). Estimates were converted to the liability scale as per ref. 45, assuming a population prevalence of 0.5%[2]. We then compared ORs per s.d. increase in PRS for each of the six PRSs using logistic regression, adjusted for age, sex and ten PCs. Finally, we added each of the six PRSs and compared the change in AUC (and 95% CI) to a baseline model comprising age, sex and the first ten PCs. The 95% CIs were computed using a stratified bootstrap with 1,000 replicates. AUCs were computed using the R package pROC[46]. The best-performing PRS (that is, the highest proportion of variance explained and change in AUC) was taken forward in downstream analyses described below.

### PRS and disease progression
We evaluated whether the PRS could aid in identifying individuals who are more likely to progress from one hepatic disease state to another. We used the following two models to evaluate disease progression: (1) from NAFLD to cirrhosis and (2) from cirrhosis to HCC. For each model, we estimated 10-year risks using Fine–Gray regression, which accounts for the competing risk of death from all causes[47]. Time zero corresponded to the first occurrence of the exposure, and individual follow-up time ended in case of the event of interest, death or end of follow-up. The earliest start of follow-up began after the time of enrollment to prevent immortal time bias. The PRS was evaluated in the UKB and validated in both CHB (NAFLD to cirrhosis and cirrhosis to HCC) and MVP (NAFLD to cirrhosis). To avoid overfitting, effects were derived from a meta-analysis that did not include the test dataset.

### Reporting summary
Further information on research design is available in the Nature Portfolio Reporting Summary linked to this article.

### Data availability
GWAS meta-analysis summary statistics are available at the GWAS Catalog (https://www.ebi.ac.uk/gwas/) (GCST90319877 and GCST90319878). The cirrhosis PRS are available at the PGS Catalog (https://www.pgscatalog.org/; PGS004621). Data from the UKB samples are available through UKB (https://www.ukbiobank.ac.uk/). FinnGen GWAS summary statistics are publicly accessible following registration (https://www.finngen.fi/en/access_results). German/UK cirrhosis cohort can be accessed at http://gengastro.med.tu-dresden.de/suppl/alc_cirrhosis/. Summary statistics from Biobank Japan are available at https://pheweb.jp/. Other individual summary statistics will be made available upon request to study PIs (AllOfUs: S.V., CHB-CID/DBDS: J.G./S.S., deCODE and Intermountain Healthcare: G.S./H.H., Estonian Biobank: T.L.). The GTEx v.8 eQTL data used in this study are available in the GTEx Portal (https://gtexportal.org/home/datasets).

### Code availability
The following software and packages were used for data analysis: PLINK 2.0 (https://www.cog-genomics.org/plink/2.0/), METAL v.2011-03-25 (http://csg.sph.umich.edu/abecasis/Metal/download/), MAGMA v.1.07 (https://ctg.cncr.nl/software/magma), EasyQC v.9.2 (https://www.uni-regensburg.de/medizin/epidemiologie-praeventivmedizin/genetische-epidemiologie/software/), LDSC v.1.0.1 (https://github.com/bulik/ldsc), PoPS v.0.1 (https://github.com/FinucaneLab/pops/tree/add-license-1), PRS-CS v.2021-06-04 (https://github.com/getian107/PRScs/), REGENIE v.2.0.1 (https://rgcgithub.github.io/regenie/), TwoSampleMR v.0.5.6 (https://mrcieu.github.io/TwoSampleMR/), MVMR 0.4 (https://github.com/WSpiller/MVMR), pROC 1.18.4 (https://cran.r-project.org/web/packages/pROC/index.html) and R v.4.1.2 (https://www.r-project.org/).

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

## Acknowledgements

This research has been conducted using the UKB resource under application 43247. This work was supported by BRIDGE—Translational Excellence Program (NNF20SA0064340 to J.G.), Beckett Fonden (23-2-10636 to J.G.), Independent Research Fund Denmark (Sapere Aude Research Leader, 9060-00012B to S.S.), Borregaard Clinical Ascending Investigator (NNF22OC0075038 to S.S.), Hallas-Møller Emerging Investigator (Novo Nordisk Foundation; NNF17OC0031204 to M.S.O.), The John and Birthe Meyer Foundation (to M.S.O.), The Innovation Fund Denmark (PM Heart to H.B.), NordForsk (to H.B.), Villadsen Family Foundation (to H.B.) and The Arvid Nilsson Foundation. Research in the MVP, funded by the Office of Research and Development, US Veterans Health Administration was supported by award MVP000 and additional funding from the Department of Veterans Affairs (award I01 BX003362 to K.-M.C.). This publication does not represent the views of the Department of Veterans Affairs, the US Food and Drug Administration or the US Government. Novo Nordisk Foundation (grants NNF17OC0027594 and NNF14CC0001 to K.B. and S.B.). The All of Us is supported by the National Institutes of Health, Office of the Director—Regional Medical Centers (1 OT2 OD026549, 1 OT2 OD026554, 1 OT2 OD026557, 1 OT2 OD026556, 1 OT2 OD026550, 1 OT2 OD 026552, 1 OT2 OD026553, 1 OT2 OD026548, 1 OT2 OD026551 and 1 OT2 OD026555), IAA (AOD 16037), Federally Qualified Health Centers (HHSN 263201600085U), Data and Research Center (5 U2C OD023196), Biobank (1 U24 OD023121), The Participant Center (U24 OD023176), Participant Technology Systems Center (1 U24 OD023163), Communications and Engagement (3 OT2 OD023205 and 3 OT2 OD023206) and Community Partners (1 OT2 OD025277, 3 OT2 OD025315, 1 OT2 OD025337 and 1 OT2 OD025276). In addition, the All of Us Research Program would not be possible without the partnership of its participants. The research was also supported by NIH HHS under grant 5T32GM136651-03 (to J.B.). This work was also supported by the NIH/NIDDK (K08 DK113109 and R01 DK131033-01A1 to S.V., and R01 DK134575 and I01 BX003362 to M.V.) and the Doris Duke Charitable Foundation (grant 2019081 to S.V.). The Estonian Biobank was funded by the European Union through the European Regional Development Fund (project 2014-2020.4.01.15-0012 GENTRANSMED) and by the Estonian Research Council (grant PRG1911). Computations were performed in the High-Performance Computing Center, University of Tartu. JK and Dallas Liver cohort are supported by NIH/NIDDK (DK090066). Finally, we would like to acknowledge the participants and investigators of the FinnGen study.

## Author contributions

J.G., G.S., H.H., H.B. and S.S. conceived the study. J.G., G.S., M.V., G.A., H.G.K., J.B., T.L., E.F., D.F.G. and S.S. performed analyses in the respective cohorts. H.H., S.V., T.L., K.-M.C., H.B. and S.S. supervised analyses in their respective cohorts. J.G., G.S., G.A., H.H., H.B. and S.S. contributed to writing the manuscript. J.G. performed meta-analysis and created figures and tables. J.G., G.S. and S.S. performed downstream analyses and drafted the manuscript. J.G., G.A., M.V., A.S., H.G.K., G.A., V.T., S.A.R., J.B., S.V., P.R.L., E.S., C.E., M.T.B., B.A.J., S.B., K.B., H.U., T.L., R.M., J.K., A.T.H., D.J., K.U.K., L.N., E.F., E.S.B., M.U., A.S., P.S., O.B.P., S.R.O., D.F.G., K.S., M.S.O., K.-M.C., H.H., H.B. and S.S. interpreted the results, reviewed and commented on the manuscript.

## Competing interests

The authors who are affiliated with deCODE genetics/Amgen declare competing financial interests as employees. H.B. receives lecture fees from Bristol-Myers Squibb, Merck Sharp and Dohme. J.G. has received lecture fee from Illumina. S.B. is a board member for Proscion A/S and Intomics A/S. N.V., L.L. and A.B. are employees at Regeneron Genetics Center. All other authors have no conflict of interest to declare.

## Additional information

**Correspondence and requests for materials** should be addressed to Jonas Ghouse or Stefan Stender.

# Reporting Summary

## Statistics

For all statistical analyses, confirm that the following items are present in the figure legend, table legend, main text, or Methods section.

| n/a | Confirmed | |
|---|---|---|
| ☐ | ☒ | The exact sample size (*n*) for each experimental group/condition, given as a discrete number and unit of measurement |
| ☒ | ☐ | A statement on whether measurements were taken from distinct samples or whether the same sample was measured repeatedly |
| ☐ | ☒ | The statistical test(s) used AND whether they are one- or two-sided *Only common tests should be described solely by name; describe more complex techniques in the Methods section.* |
| ☐ | ☒ | A description of all covariates tested |
| ☐ | ☒ | A description of any assumptions or corrections, such as tests of normality and adjustment for multiple comparisons |
| ☐ | ☒ | A full description of the statistical parameters including central tendency (e.g. means) or other basic estimates (e.g. regression coefficient) AND variation (e.g. standard deviation) or associated estimates of uncertainty (e.g. confidence intervals) |
| ☐ | ☒ | For null hypothesis testing, the test statistic (e.g. $F$, $t$, $r$) with confidence intervals, effect sizes, degrees of freedom and $P$ value noted *Give P values as exact values whenever suitable.* |
| ☒ | ☐ | For Bayesian analysis, information on the choice of priors and Markov chain Monte Carlo settings |
| ☒ | ☐ | For hierarchical and complex designs, identification of the appropriate level for tests and full reporting of outcomes |
| ☐ | ☒ | Estimates of effect sizes (e.g. Cohen's *d*, Pearson's *r*), indicating how they were calculated |

*Our web collection on statistics for biologists contains articles on many of the points above.*

## Software and code

Policy information about availability of computer code

| Data collection | No software was used for data collection. |
|---|---|
| Data analysis | The following software and packages were used for data analysis: PLINK 2.0 (https://www.cog-genomics.org/plink/2.0/), METAL v.2011-03-25 (http://csg.sph.umich.edu/abecasis/Metal/download/), MAGMA v.1.07 (https://ctg.cncr.nl/software/magma), EasyQC v.9.2 (https://www.uni-regensburg.de/medizin/epidemiologie-praeventivmedizin/genetische-epidemiologie/software/), LD score regression v.1.0.1 (https://github.com/bulik/ldsc), PoPS v.0.1 (https://github.com/FinucaneLab/pops/tree/add-license-1), PRS-CS v.2021-06-04 (https://github.com/getian107/PRScs/), REGENIE v.2.0.1 (https://rgcgithub.github.io/regenie/), TwoSampleMR v.0.5.6 (https://mrcieu.github.io/TwoSampleMR/), MVMR 0.4 (https://github.com/WSpiller/MVMR), pROC 1.18.4 (https://cran.r-project.org/web/packages/pROC/index.html) and Rv.4.1.2 (https://www.r-project.org/). |

For manuscripts utilizing custom algorithms or software that are central to the research but not yet described in published literature, software must be made available to editors and reviewers. We strongly encourage code deposition in a community repository (e.g. GitHub). See the Nature Portfolio guidelines for submitting code & software for further information.

## Data

Policy information about availability of data

All manuscripts must include a data availability statement. This statement should provide the following information, where applicable:
  - Accession codes, unique identifiers, or web links for publicly available datasets
  - A description of any restrictions on data availability
  - For clinical datasets or third party data, please ensure that the statement adheres to our policy

GWAS meta-analysis summary statistics are available at GWAS Catalogue (GCST90319877 and GCST90319878). The cirrhosis PRS are available at PGS Catalog (PGS004621). Individual-level data sharing is subject to restrictions imposed by patient consent and local ethics review boards. The GTEx v.8 eQTL data used in this study are available in the GTEx Portal (https://gtexportal.org/home/datasets).

## Human research participants

Policy information about studies involving human research participants and Sex and Gender in Research.

| | |
|---|---|
| Reporting on sex and gender | We used biological sex throughout. |
| Population characteristics | Cohort characteristics (age and sex distribution) are provided in Supplementary Table 1 and each cohort has been described in more detail in Supplementary Information. |
| Recruitment | Recruitment information is provided in Supplementary Information. |
| Ethics oversight | All human research was approved within each contributing study by the relevant institutional review board and conducted according to the Declaration of Helsinki. CHB-CID/DBDS: National Committee on Health Research Ethics; deCODE: National Bioethics Committee; Intermountain Healthcare: Intermountain Healthcare Institutional Review Board; UKB: Northwest Multicenter Research Ethics Committee; Geisinger DiscovEHR: The GHS project has received ethical approval from the Geisinger Health System Institutional Review Board under project no. 2006-0258; FinnGen: The Coordinating Ethics Committee of the Hospital District of Helsinki and Uusimaa; Estonian Biobank: Ethical approval 1.1-12/624 from the Estonian Committee on Bioethics and Human Research, Estonian Ministry of Social Affairs; Biobank Japan: Research ethics committees at the Institute of Medical Science, the University of Tokyo, the RIKEN Yokohama Institute, and the 12 cooperating hospitals; Copenhagen General Population Study and Copenhagen City Heart Study: institutional review boards and Danish ethical committees; All of Us: National Institute of Health All of Us Institutional Review Board; and Dallas Liver Cohort: University of Texas Southwestern Institutional Review Board. MVP: VA Central Institutional Review Board (IRB). All participants (except for CHB-CID) provided written informed consent. For CHB-CID, patients were informed about the opt-out possibility of having their biological specimens excluded from use in research in general. Since 2004, a national Register on Tissue Application (Vævsanvendelsesregistret) lists all individuals who have chosen to opt out and whose samples cannot be used for research purposes. Before initiating this study, individuals listed in the Register on Tissue Application were excluded. |

Note that full information on the approval of the study protocol must also be provided in the manuscript.

# Field-specific reporting

Please select the one below that is the best fit for your research. If you are not sure, read the appropriate sections before making your selection.

☒ Life sciences  ☐ Behavioural & social sciences  ☐ Ecological, evolutionary & environmental sciences

For a reference copy of the document with all sections, see nature.com/documents/nr-reporting-summary-flat.pdf

# Life sciences study design

All studies must disclose on these points even when the disclosure is negative.

| | |
|---|---|
| Sample size | All available samples passing QC were used to maximize power. |
| Data exclusions | Within each contributing study, samples were excluded on the basis of well-established individual and variant quality control procedures to remove poor quality genotypes, SNPs and samples. Quality control filters are provided in Supplementary Table 1 and in under Methods. |
| Replication | We replicated significant variants once in the MVP cohort |
| Randomization | Randomization is not relevant since this is a retrospective case-control study and there was no treatment to randomize. |
| Blinding | Blinding was not relevant because it is a retrospective case-control study and there was no randomized allocation to be blinded to. |

# Reporting for specific materials, systems and methods

We require information from authors about some types of materials, experimental systems and methods used in many studies. Here, indicate whether each material, system or method listed is relevant to your study. If you are not sure if a list item applies to your research, read the appropriate section before selecting a response.

## Materials & experimental systems

| n/a | Involved in the study |
|-----|----------------------|
| ☒ ☐ | Antibodies |
| ☒ ☐ | Eukaryotic cell lines |
| ☒ ☐ | Palaeontology and archaeology |
| ☒ ☐ | Animals and other organisms |
| ☒ ☐ | Clinical data |
| ☒ ☐ | Dual use research of concern |

## Methods

| n/a | Involved in the study |
|-----|----------------------|
| ☒ ☐ | ChIP-seq |
| ☒ ☐ | Flow cytometry |
| ☒ ☐ | MRI-based neuroimaging |

