## [Peer Review File · Nature Genetics]

Peer Review Information

Manuscript Title: Integrative common and rare variant analyses provide insights into the genetic architecture of liver cirrhosis

Corresponding author name(s): Dr Jonas Ghouse

Reviewer Comments & Decisions:

Decision Letter, initial version:

8th Aug 2023

Dear Dr Ghouse,

Your Article, "Integrative common and rare variant analyses provide insights into the genetic architecture of liver cirrhosis" has now been seen by 3 referees. You will see from their comments copied below that while they find your work of considerable potential interest, they have raised quite substantial concerns that must be addressed. In light of these comments, we cannot accept the manuscript for publication, but would be interested in considering a substantially revised version that addresses these serious concerns.

We hope you will find the referees' comments useful as you decide how to proceed. If you wish to submit a substantially revised manuscript, please bear in mind that we will be reluctant to approach the referees again in the absence of major revisions.

To guide the scope of the revisions, the editors discuss the referee reports in detail within the team with a view to identifying key priorities that should be addressed in revision. As you will see from these comments, Reviewer #1 has major concerns about the study design and the rigor of the statistical analyses; Reviewer #2 notes that there is an inherent challenge in interpreting case-control GWAS for cirrhosis and downstream analyses such as PheWAS, MR, and gene prioritization. We ask that you strengthen the analyses as suggested, improve the discussion, and address all referee comments as thoroughly as possible with appropriate revisions. We hope that you will find the prioritized set of referee points to be useful when revising your study. Please do not hesitate to get in touch if you would like to discuss these issues further.

If you choose to revise your manuscript taking into account all reviewer and editor comments, please highlight all changes in the manuscript text file. At this stage we will need you to upload a copy of the manuscript in MS Word .docx or similar editable format.

*2) If you have not done so already please begin to revise your manuscript so that it conforms to our Article format instructions, available here. Refer also to any guidelines provided in this letter.

Please be aware of our guidelines on digital image standards.

[redacted]

If you wish to submit a suitably revised manuscript we would hope to receive it within 6 months. If you cannot send it within this time, please let us know. We will be happy to consider your revision so long as nothing similar has been accepted for publication at Nature Genetics or published elsewhere. Should your manuscript be substantially delayed without notifying us in advance and your article is eventually published, the received date would be that of the revised, not the original, version.

Thank you for the opportunity to review your work.

Sincerely,
Wei

Wei Li, PhD
Senior Editor
Nature Genetics
New York, NY 10004, USA
www.nature.com/ng

Reviewers' Comments:

Reviewer #1:

Remarks to the Author:

This is an interesting study by Ghouse et al. reporting 36 variants and other insight into genetics of liver cirrhosis using integrative common and rare variant analyses mostly in European individuals. While the results are of interest in elucidating this important liver disease, cirrhosis, I have several concerns about the study design and the rigor of the statistical analysis that somewhat lessen my enthusiasm.

1. The study design is predominantly European (87.9%), preventing powerful enough cirrhosis analysis of other populations. No replications are conducted for the cirrhosis results, which decreases my enthusiasm especially due to the issues related to the study design described in point 2.

2. Of the 36 variants, only 15 (41.7%) variants were identified to be genome-wide significant in the actual cirrhosis GWAS, whereas 21 (58.3%) variants were identified in the GWAS of liver enzyme scans of alanine aminotransferase and gamma-glutamyl transferase with much weaker significance for cirrhosis (FDR<5%). Thus, these 21 variants are not genome-wide significant for cirrhosis, and despite this limitation, no replication analyses were conducted. The rationale of this design is unclear and replications of the weak cirrhosis variants are warranted before drawing these types of strong conclusions of their role in cirrhosis. It is also problematic for evaluation of the results that the actual FDR values of the 21 variants are not given in Suppl Table 8 or elsewhere in the manuscript.

3. Due to the atypical GWAS significance criteria (i.e. FDR<5%) used for the 21 SNPs, it would be important to estimate how much the 21 variants contribute to the European PRS of cirrhosis (using permutations of the same number of random clumped SNPs) to investigate whether a significant amount of trait variation is explained by these SNPs when compared to random sets of 21 clumped SNPs. The author should also compare the cirrhosis PRS obtained with the 21 variants to the PRS obtained using the overall 36 SNPs and the 15 genome-wide significant cirrhosis GWAS SNPs to assess whether the 21 SNPs significantly contribute to cirrhosis PRS.

4. The study design of the gene prioritization of the variants is not well justified. No focus on liver expression is set, as is exemplified by the result that only 2 of the 104 cis-eQTLs are identified in the liver, as seen in Suppl Table 14. The rationale for the eQTL and sQTL design that does not focus on the liver should be clarified. The differences in the sample sizes of eQTL data sets across the tissues between GTEx and deCODE, and even across the GTEx tissues, are large, and this discrepancy and its effects on power to detect eQTLs has not been discussed.

5. Formal colocalization analyses should be conducted for the GWAS and eQTL overlap analyses to check which GWAS variants affect local gene expression.

6. In the PRS and disease progression analyses, there are very few individuals developing HCC (n=91) and cirrhosis (n=193) during the follow-up time in the UK Biobank. Thus, the results of Figure 4b-c should be replicated before drawing these types of strong conclusions in the Abstract and elsewhere in the manuscript. Overall, the current strong conclusions of these PRS results are not well-justified due to the small sample sizes and missing replications.

7. Numbers of individuals included in the analyses should be added to Figures 2-4 to increase the scientific rigor and robustness of the manuscript. Similarly, the investigated trait should be mentioned in the titles of Suppl tables 2-4. Now these Suppl tables do not mention which trait was analyzed.

8. The authors refer to the effect of GPAM loss-of-function variants on cirrhosis as a trend both in the Results and Discussion; however, the p-value for this result is p=0.296, which is not a trend. This should be corrected.

Reviewer #2:

Remarks to the Author:

This study builds upon a GWAS of 18,265 liver cirrhosis cases and 1,782,047 controls, thereby increasing the case sample size for this trait by over three times. Compared to the previous work of Emdin et al., data from the CHB-CID/DBDS, Geisinger DiscovEHR, Intermountain Healthcare, All of Us, deCODE, and the Estonian Biobank were incorporated, whereas BioVU, ERIC, and Partners Biobank were omitted from this study. The authors identified 15 genetic loci associated with liver cirrhosis, of which eight are novel. Four of the previously reported nine loci showed nominal significance in their data set. They further added 21 ALT/GGT signals which also showed marginal associations with cirrhosis. Subsequently, they conducted downstream and PRS analyses. Rare

variant analyses were also performed using exome sequencing data.

However, I believe there is an inherent challenge in interpreting case-control GWAS for cirrhosis. The cirrhosis case samples are disproportionately patients with alcoholic and non-alcoholic fatty liver diseases, viral hepatitis, and genetic liver diseases. Certain genetic variants may influence the progression of liver inflammation and/or fibrosis, some may contribute to both the underlying diseases and cirrhosis, and others may only represent susceptibility variants for underlying diseases without direct impact on cirrhosis development. A standard GWAS, which compares allele frequency differences between cases and controls, may not distinguish among these scenarios, complicating the interpretation of downstream analyses such as PheWAS, MR, and gene prioritization. For instance, PNPLA, TM6SF2, and MTARC1 are well-known NAFLD loci. Is it possible to discern if these loci showed association merely because NAFLD is common among cirrhosis patients, or do these loci directly influence cirrhosis development? If they are merely NAFLD variants, the value of downstream analysis such as gene prioritization in cirrhosis research is limited. Also, does the HLA locus only identified in cross-ancestry GWAS indicate viral hepatitis locus, or does it truly affect developing cirrhosis?

The authors identified 15 significant cirrhosis signals and included an additional 21 signals, selected from SNPs significantly associated with ALT or GGT. Their p-values for cirrhosis showed a false discovery rate (FDR) of less than 0.05. However, this approach appears operationally influenced. I believe such results would benefit from replication for the target trait, following the approach used in the Emdin et al. paper. Also I think it might be more natural to conduct this analysis using MTAG.

The interpretation of MR result is further challenging. As many cirrhosis loci are shared with NAFLD, do the causal roles of BMI and alcohol intake for cirrhosis or for NAFLD? Can the authors clarify them?

The interpretation of the MR results presents additional challenges. Given that many cirrhosis loci are shared with NAFLD, it becomes difficult to discern whether the causal roles of BMI and alcohol intake are directly causal for cirrhosis or for NAFLD. I would appreciate if the authors could provide some discussion.

Regarding the gene burden test conducted on 19 genes, I propose a subtitle change for this section, as the standard goal of a rare variant burden test might be to identify genes not detected by the common variant association test. These 19 genes were already identified. By the way, it would be helpful if the authors could clarify their selection criteria. The analysis provides an interesting interpretation of the protective or risk effects of each gene variant. If this is the focus, it might be more beneficial to evaluate not only these 19 genes but also other genes within the common variant association loci that are in LD with GWAS lead variants, for a fine mapping purpose.

Even with the complications posed by cirrhosis GWAS as mentioned above, the construction of a PRS is meaningful if validated by an independent sample set. The PRS in this study was constructed without the UK dataset and later evaluated in the UKB dataset, thus making the evaluation scheme valid. The performance of the cirrhosis-PRS was superior when compared to the ALT-PRS. Was the improved performance in the AUC statistically significant? The authors could also evaluate the NRI. Lastly, the reported higher risk of developing HCC among cirrhosis patients and developing cirrhosis among NAFLD patients remain significant findings.

Minor points:

Several technical comments on the MR analysis:

- To minimize weak instrumental bias, the authors excluded traits with fewer than ten instrumental

variables. I recommend performing an F-test to further mitigate this issue.

- One potential violation of MR assumptions is linkage disequilibrium (LD). The authors considered r^2 in clumping to resolve this issue but did not specify the source of their r^2 data. This should be clarified.
- There might be a potential bias introduced by using outcome summary statistics with LD structures slightly different from European ancestry such as Biobank Japan, All of Us (Hispanic), and All of Us (African American). I encourage a discussion on possible biases from using these datasets, or the addition of some sensitivity analyses. Please consider using the same single-ancestry derived GWAS for exposure and outcome.
- The data sources for exposure body mass index (BMI) and alcohol were not provided. Please clarify these and their source populations.
- In the methods section, the authors mentioned, "We used two different MR methods: inverse variance weighted (IVW) model as our primary model and the weighted median model as sensitivity analysis. MR-Egger-intercept was used to for pleiotropy." However, I noticed results from MR-PRESSO in Supplementary Figures 1 and 2. This discrepancy needs to be resolved.

Additional comments:

- For SNP heritability analysis, a clearer description of the method would be beneficial, instead of stating "default settings".
- The authors found a significant gene-environment interaction of the PNPLA3 variant with alcohol intake, obesity, and T2D. The validity of testing for a gene-environment interaction term depends on the absence of a gene-environment association (Dudbridge, AJHG 2014). While I do not believe there is a strong association of this gene with these environmental factors, I recommend stating this more clearly in the text.

Reviewer #3:

Remarks to the Author:

The authors perform the to date largest GWAS for the liver cirrhosis phenotype harnessing multiple international cohorts. The study is timely and well performed and will be met with substantial interest in the liver community.

I have only a couple stylistic / minor requests:

- 1) A traditional table with the identified loci, combined p-value and OR would be nice as part of the main manuscript.
- 2) A stronger focus on differentiation of alcoholic and non-alcoholic liver disease (are they really different?) would be helpful.
- 3) Supplementary tables should be formatted in landscape mode to fit on one page.

Author Rebuttal to Initial comments

Manuscript ID: NG-A62855.R1

Manuscript title: Integrative common and rare variant analyses provide insights into the genetic architecture of liver cirrhosis

Point-by-point responses to comments from Reviewer #1.

This is an interesting study by Ghouse et al. reporting 36 variants and other insight into genetics of liver cirrhosis using integrative common and rare variant analyses mostly in European individuals. While the results are of interest in elucidating this important liver disease, cirrhosis, I have several concerns about the study design and the rigor of the statistical analysis that somewhat lessen my enthusiasm.

Response: Thank you for the careful review of our work, and for providing constructive comments. Our responses are provided in blue below. Text incorporated into the revised paper is shown in red.

1. The study design is predominantly European (87.9%), preventing powerful enough cirrhosis analysis of other populations. No replications are conducted for the cirrhosis results, which decreases my enthusiasm especially due to the issues related to the study design described in point 2.

Response: We agree that independent replication would strengthen the study, as also pointed out by Reviewer #2 (comment #2). We have therefore now performed a validation analysis in the Million Veteran Program (MVP) cohort which includes 21,689 cirrhosis cases and 617,729 controls. Of the 36 variants associated with cirrhosis in our original study, 20 associated with cirrhosis at $P < 0.05$ and consistent direction of effect in the MVP cohort. Of the 15 variants that did not replicate at $P < 0.05$, we found a high level of concordance in the directions of effects between the two datasets.

Please note that the Asian-specific risk variant in *ALDH2* was not amenable to replication owing to the low number of participants of East Asian ancestry in the MVP cohort. However, the association of this variant with reduced alcohol intake in Asian populations has been firmly established since the 1980's. The variant has also been associated with protection from alcohol-related liver disease in numerous studies (e.g. PMIDs: 7904979, 29779728, 11748356, 37378938, 27189280). These associations mitigate the need for replication of its association with cirrhosis. Moreover, the *HLA* variant was unlikely to replicate in MVP given that its association with cirrhosis in our GWAS was entirely driven by Biobank Japan. The association of this variant with cirrhosis is likely secondary to its association with risk of chronic hepatitis in Asian populations (please also see our response to comment #1 from Reviewer #2 regarding the *HLA* variant).

The validation data have been incorporated into the revised manuscript as follows:

Abstract, Page 3, Line 2: 'Using data from 12 cohorts, including 18,265 cases with cirrhosis, 1,782,047 controls, up to 1 million individuals with liver function tests, and a validation cohort of 21,689 cases and 617,729 controls, we identify and validate 20 risk associations for liver cirrhosis.'

Methods, Page 16, Line 1: "**Validation.** To validate our findings, we performed replication of cirrhosis variants identified via cirrhosis GWAS and/or endophenotype-informed analysis using summary statistics of 21,689 cirrhosis cases and 617,729 disease-free controls from the MVP. Two variants (rs146650659 and rs113469203) were not available in the MVP, for which we selected

suitable proxies ($r^2 \geq 0.65$). rs671 in *ALDH2* was not amenable to validation, due to low frequency in non-East Asian populations. Cases were defined as in the primary GWAS analysis (Supplementary Table 1). A $P < 0.05$ and consistent direction of effect was considered successful replication.”

Results, Page 6, Line 11: “**Validation.** A total of 36 risk variants were identified through cirrhosis GWAS and/or the endophenotype-informed analysis, of which 35 were available for replication in the Million Veterans Program (MVP) cohort (21,689 cases and 617,729 controls). Replication of *ALDH2* p.Glu504Lys (rs671) was not possible due to absence in non-East Asian populations. Twenty variants (57 %) showed significant associations ($P < 0.05$) with cirrhosis and concordant directions of effect in MVP (**Supplementary Table 10**). Ten of the replicated variants were initially identified in stage 1 or 2 of the GWAS, whereas 10 were identified solely via endophenotype-informed analyses. Of the 15 variants that did not replicate at $P < 0.05$, we found a high level of concordance in the magnitudes and directions of effects between the two datasets (Pearson’s $r^2 = 0.73$, $P = 1.8 \times 10^{-3}$; **Supplementary Table 10**).”

Discussion, Page 12, Line 3: “Our study included over 18,000 cirrhosis cases and more than one million individuals with endophenotypic data sampled from four populations and identified 36 risk variants for cirrhosis of which 20 replicated in an independent cohort.”

The validation results are shown in a new **Supplementary Table 10**.

2a. *Of the 36 variants, only 15 (41.7%) variants were identified to be genome-wide significant in the actual cirrhosis GWAS, whereas 21 (58.3%) variants were identified in the GWAS of liver enzyme scans of alanine aminotransferase and gamma-glutamyl transferase with much weaker significance for cirrhosis (FDR<5%). Thus, these 21 variants are not genome-wide significant for cirrhosis, and despite this limitation, no replication analyses were conducted. The rationale of this design is unclear and replications of the weak cirrhosis variants are warranted before drawing these types of strong conclusions of their role in cirrhosis.*

Response: Please see our response to comment #1 above. We have now performed replication analyses in the MVP. Of the 21 variants identified solely through endophenotype-driven analyses, 10 associated with cirrhosis ($P < 0.05$) with concordant direction of effect in the MVP cohort.

Results, Page 6, Line 15: “Ten of the replicated variants were initially identified in stage 1 or 2 of the GWAS, whereas 10 were identified solely via endophenotype-informed analyses. Of the 15 variants that did not replicate at $P < 0.05$, we found a high level of concordance in the magnitudes and directions of effects between the two datasets (Pearson’s $r^2 = 0.73$, $P = 1.8 \times 10^{-3}$; **Supplementary Table 10**).”

2b. It is also problematic for evaluation of the results that the actual FDR values of the 21 variants are not given in Suppl Table 8 or elsewhere in the manuscript.

Response: We have now added FDR values of the variants in **Supplementary Table 8**.

3a. Due to the atypical GWAS significance criteria (i.e. FDR<5%) used for the 21 SNPs, it would be important to estimate how much the 21 variants contribute to the European PRS of cirrhosis (using permutations of the same number of random clumped SNPs) to investigate whether a significant amount of trait variation is explained by these SNPs when compared to random sets of 21 clumped SNPs.

Response: We now report the associations and trait variance explained by a PRS based on these 21 variants, both in UK Biobank and MVP. We have also updated **Supplementary Table 9**, which now includes associations between PRS_{21-SNP} and hepatobiliary outcomes.

Results Page 6, Line 7: “We found that a PRS using these 21 variants identified via endophenotype-driven analysis associated significantly with cirrhosis in the UKB (OR 1.15 per s.d., 95% CI 1.11-1.20, $P = 1.8 \times 10^{-11}$) and MVP (OR 1.09 per s.d., 95% CI 1.07-1.11, 1.20, $P = 3.6 \times 10^{-19}$), but contributed only little to the variance explained ($r^2_{UKB} = 0.2\%$ and $r^2_{MVP} = 0.1\%$, **Supplementary Table 9**).“

We also assessed the association of a PRS based on a set of 21 random SNPs, with weights derived from the primary GWAS excluding the UKB. We tested for association in both the UKB and MVP cohorts. Notably, this PRS did not associate with cirrhosis in either the UKB ($P = 0.387$) or MVP ($P=0.67$) and did not account for any variance in the trait ($r^2=0.0\%$ in both cohorts). We have chosen not to include these data in the revised paper as we view them as tangential and likely to confuse the reader, especially in light of the multitude of other PRSs presented. Of note, we now show results for both a 15-SNP PRS and a 36-SNP PRS, highlighting the incremental explanatory power in the 21 endophenotype-identified SNPs, as well as data pertaining to the 21-endophenotype-derived-SNP PRS, as outlined above.

3b. The author should also compare the cirrhosis PRS obtained with the 21 variants to the PRS obtained using the overall 36 SNPs and the 15 genome-wide significant cirrhosis GWAS SNPs to assess whether the 21 SNPs significantly contribute to cirrhosis PRS.

Response: We appreciate the Reviewer’s comment and agree that investigating the amount of trait variance explained by the 21 variants compared to the other PRS is valuable. These results are now shown and discussed. In brief, a PRS based on the 21 SNPs was significantly associated with cirrhosis but explained relatively little trait variation. Please see the following (in particular Supplementary Tables 9 and 24):

Results, Page 10, Line 4: “To explore this, we performed a range of analyses in the UKB. We created six distinct PRSs - a European-specific (PRS_{EUR}), a cross-ancestry PRS (PRS_{CA}), a PRS based on ALT (PRS_{ALT}), and three different weighted scores, each incorporating varying numbers of risk variants identified in this study. We then compared the predictive ability of each of these PRSs. We found that the PRS_{15-SNP} explained the highest proportion of phenotypic variation ($r^2 = 1.7\%$; **Supplementary Table 9**), change in AUC (+0.031, 95% CI 0.023-0.039; **Supplementary Table 24**) and yielded an OR for cirrhosis of 1.42 per s.d. increase in PRS (**Supplementary Table 9** and **Figure 5A**). In comparison, the PRS_{ALT} accounted for 1.3% of cirrhosis phenotypic variance, had a change in AUC of 0.021 and an OR of 1.38 per s.d. increase in PRS (**Figure 5**). The difference in predictive ability between PRS_{15-SNP} and PRS_{ALT} was statistically significant (change in AUC +0.005, 95% CI 0.003-0.017, $P = 0.005$). Next, we evaluated reclassification of individuals after addition of the PRS_{15-SNP} to a baseline model containing age, sex, and 10 PCs. Adding PRS_{15-SNP} resulted in a net percentage of individuals with cirrhosis correctly classified upward (event net reclassification index [NRI]) of 8.4% (95% CI 3.1–13.7), and of individuals without cirrhosis correctly classified downward (non-event NRI) of 21.3% (95% CI 20.1–22.7). These changes resulted in an overall continuous NRI of 29.7% (95% CI 23.4–36.1). Following this, we investigated how the various PRSs associated with a broader range of hepatobiliary outcomes. We found that the PRS_{EUR} had the highest OR for HCC, for which a 1 s.d. higher PRS conferred an OR of 1.67 (95% CI 1.52-1.82), followed by liver-related death (OR 1.56 [95% CI, 1.44-1.69]) and alcoholic cirrhosis (OR 1.47 [95% CI 1.39-1.57]). Across the range of outcomes, the PRS_{15-SNP} tended to have slightly larger per s.d. effect sizes than PRS_{CA}, PRS_{ALT}, and PRS_{5-SNP}, but comparable to the PRS_{EUR}.”

4. The study design of the gene prioritization of the variants is not well justified. No focus on liver expression is set, as is exemplified by the result that only 2 of the 104 cis-eQTLs are identified in the liver, as seen in Suppl Table 14. The rationale for the eQTL and sQTL design that does not focus on the liver should be clarified. The differences in the sample sizes of eQTL data sets across the tissues between GTEx and deCODE, and even across the GTEx tissues, are large, and this discrepancy and its effects on power to detect eQTLs has not been discussed.

Response: We acknowledge the importance of liver-specific expression data for understanding of the variant’s impact on liver pathology. However, obtaining sizable liver expression datasets coupled with genotype information specifically from liver tissue is difficult, and at present, the only publicly available dataset for liver expression comes from GTEx with a sample size of N=226. While we acknowledge the benefit of concentrating solely on liver tissue, we also recognize that many

candidate genes exhibit expression across multiple tissues beyond the liver. Therefore, we made the decision to explore expression datasets from other tissues, such as blood, where larger sample sizes are available. This approach allows us to increase statistical power for detecting associations between genetic variants and expression patterns. We have however highlighted that two genes showed evidence for colocalization in liver tissue. Please see changes below:

Results, Page 8, Line 30: “Only *MBOAT7* and *HKDC1* showed evidence of colocalization in liver tissue.”

Discussion, Page 14, Line 8: “Third, although we included liver tissue in our eQTL analyses, the majority of eQTLs that we report are based on datasets from non-hepatic tissues, some of which had manyfold larger sample sizes. This limits the ability to draw conclusions on liver expression specificity of the reported loci. Nevertheless, we did not solely depend on eQTL signals as standalone evidence but reported potential effector genes when complementary evidence from other gene mapping strategies converged on the same gene.”

5. Formal colocalization analyses should be conducted for the GWAS and eQTL overlap analyses to check which GWAS variants affect local gene expression.

Response: Thank you for this comment. We have now formally tested for colocalization, prioritizing genes for which there were significant eQTL associations. In total, we found evidence for colocalization at six loci, pointing at 12 potentially causative genes. Of the six loci, colocalization prioritized a single gene at two loci (*HSD17B13* and *TOR1B*). Since our gene prioritization approach was based on eQTL associations rather than formal colocalization analysis, 18 instead of 19 genes are now supported by at least two lines of evidence. We have made the necessary changes throughout the manuscript, figures, and supplementary tables. We have added the following:

Methods, Page 18, Line 5: “We used colocalization analyses to detect shared causal variants between cirrhosis and gene expression using COLOC (v.3.2.1) R package.³⁷ We tested genes with significant *cis*-eQTL association by analyzing all variants that were located within a ± 1 -Mb window around the sentinel variant using eQTL and cirrhosis, ALT and GGT meta-analysis summary statistics. We set the prior probabilities to $p_1 = 1 \times 10^{-4}$, $p_2 = 1 \times 10^{-4}$, $p_{12} = 5 \times 10^{-6}$ as suggested previously.³⁸ We report the posterior probability that the association with gene expression and cirrhosis risk is driven by a single causal variant. We consider a PPa ≥ 0.70 as supporting evidence for a causal role for the gene as a mediator of cirrhosis.”

Results, Page 8, Line 28: “Using gene expression data, we found significant colocalization (posterior probability [PPa] > 0.70) at six loci (**Supplementary Tables 17 and 18**), proposing 12 potentially

causal genes, and a single gene at two loci (*HSD17B13* and *TOR1B*). Only *MBOAT7* and *HKDC1* showed evidence of colocalization in liver tissue.”

6a. In the PRS and disease progression analyses, there are very few individuals developing HCC (n=91) and cirrhosis (n=193) during the follow-up time in the UK Biobank. Thus, the results of Figure 4b-c should be replicated before drawing these types of strong conclusions in the Abstract and elsewhere in the manuscript.

Response: We have now repeated the PRS progression analyses in the CHB and MVP cohorts, with similar results. Please see results below and **Figure 6**. Please note that the cirrhosis to HCC analysis could not be re-tested in MVP due to lack of shareable data relating to this progression.

Methods, Page 20, Line 17: “The PRS was evaluated in the UKB and validated in both CHB (NAFLD to cirrhosis and cirrhosis to HCC) and MVP (NAFLD to cirrhosis). To avoid overfitting, effects were derived from meta-analysis that did not include the test dataset.”

Results, Page 11, Line 4: “A similar pattern was observed in CHB, involving 3,253 individuals with cirrhosis, of whom 172 developed HCC. Individuals in the top 20% of the PRS had an 11% (95% CI 8.5-14.0) risk of developing HCC, compared to 5.3% (95% CI 4.4-6.3, P for difference < 0.001) for those in the bottom 80% (**Figure 6b**). Correspondingly, the PRS associated with increased risk of progressing to cirrhosis in individuals with registry-defined NAFLD (**Figure 6**). We identified 4,449 individuals in the UKB with registry-defined NAFLD, of whom 193 progressed to cirrhosis during follow-up. Individuals with a PRS_{15-SNP} in the top 20% had a 10-year risk of 11.0% (95% CI 7.1-16.0), whereas individuals in the bottom 80% of the distribution had a 10-year risk of 8.6% (95% CI 6.8-11.0, P for difference = 0.036; **Figure 6c**). In CHB, among 860 individuals with registry-defined NAFLD, 95 developed cirrhosis during follow-up. In MVP, out of the 18,302 individuals with NAFLD, 280 developed cirrhosis. Those in the top 20% of the PRS_{15-SNP} distribution had a 10-year cirrhosis risk of 13.0% (95% CI 7.5-19.0) in CHB (**Figure 6d**), and a 5-year risk of 2.8% (95% CI 2.3-3.5) in MVP (**Figure 6e**), respectively. In contrast, those in the bottom 80% of PRS_{15-SNP} had a 10-year risk of 9.9% (95% CI 7.6-12.0, P for difference = 0.032) in CHB and a 5-year risk of 1.5% (95% CI 1.3-1.7, P for difference < 0.001) in MVP, respectively.

Figure 6. Polygenic risk scores and disease progression. a,- b, Risk of hepatocellular carcinoma (HCC) in individuals with liver cirrhosis according to polygenic risk percentile. **c,-d,** Risk of cirrhosis in individuals with non-alcoholic fatty liver disease (NAFLD) diagnosis, according to polygenic risk percentile. We used the PRS_{15-SNP} to assign individuals to different risk groups. Cumulative incidence was estimated using the Fine-Gray regression, which takes the competing risk of death into account. Number of individuals at risk according to each exposure group and events are given below each plot. UKB, UK Biobank; CHB, Copenhagen Hospital Biobank; MVP, Million Veteran Program.

6b. Overall, the current strong conclusions of these PRS results are not well-justified due to the small sample sizes and missing replications.

Response: We agree that the conclusions about clinical utility of these PRS results should be softened, even with the above-described replications included. The conclusion in the abstract has been revised accordingly:

Abstract, Page 3, Line 13: “In conclusion, this study provides new insights into the genetic underpinnings of cirrhosis.”

7. Numbers of individuals included in the analyses should be added to Figures 2-4 to increase the scientific rigor and robustness of the manuscript. Similarly, the investigated trait should be mentioned in the titles of Suppl tables 2-4. Now these Suppl tables do not mention which trait was analyzed.

Response: In response to the Reviewer’s suggestion, we have included the number of individuals included in the analyses (N_{case} for binary traits and N_{total} for quantitative traits) in **Figures 2 and 4** (now **Figure 5**) and we added a risk table displaying numbers at risk in **Figure 6**. However, we think that adding N_{exposed} and N_{event} within each subcategory in **Figure 3** (now **Figure 4**) would make the figure less readable. We realized that the number of events is currently lacking in **Supplementary Table 15**, which has now been added. Additionally, we have revised the titles of Supplementary Tables 2-4 to explicitly mention "cirrhosis", to clarify which trait is being investigated.

8. The authors refer to the effect of GPAM loss-of-function variants on cirrhosis as a trend both in the Results and Discussion; however, the p-value for this result is $p=0.296$, which is not a trend. This should be corrected.

Response: We agree with the Reviewer that this may be misleading. We have now revised the sentences in the Results and Discussion sections that referred to the effect as a trend.

Results, Page 9, Line 16: “Notably, rare pLoF variants in *GPAM* were associated with lower ALT levels (-0.29 s.d. units per allele, 95% CI -0.40 to -0.16 , $P = 5.8 \times 10^{-6}$) and numerically lower odds of cirrhosis, although the latter association did not reach statistical significance (OR 0.36, 95% CI 0.05-2.42, $P = 0.296$; **Supplementary Table 23**).”

Discussion, Page 12, Line 27: **“By extending the allelic series, we showed that rare loss-of-function variants in *GPAM* associate with lower plasma ALT levels.”**

Manuscript ID: NG-A62855.R1

Manuscript title: Integrative common and rare variant analyses provide insights into the genetic architecture of liver cirrhosis

Point-by-point responses to comments from Reviewer #2.

Remarks to the Author:

This study builds upon a GWAS of 18,265 liver cirrhosis cases and 1,782,047 controls, thereby increasing the case sample size for this trait by over three times. Compared to the previous work of Emdin et al., data from the CHB-CID/DBDS, Geisinger DiscovEHR, Intermountain Healthcare, All of Us, deCODE, and the Estonian Biobank were incorporated, whereas BioVU, ERIC, and Partners Biobank were omitted from this study. The authors identified 15 genetic loci associated with liver cirrhosis, of which eight are novel. Four of the previously reported nine loci showed nominal significance in their data set. They further added 21 ALT/GGT signals which also showed marginal associations with cirrhosis. Subsequently, they conducted downstream and PRS analyses. Rare variant analyses were also performed using exome sequencing data.

Response: Thank you for taking the time to review our work, and for providing constructive comments. Our responses are provided in blue below. Text incorporated into the revised paper is shown in red.

1. However, I believe there is an inherent challenge in interpreting case-control GWAS for cirrhosis. The cirrhosis case samples are disproportionately patients with alcoholic and non-alcoholic fatty liver diseases, viral hepatitis, and genetic liver diseases. Certain genetic variants may influence the progression of liver inflammation and/or fibrosis, some may contribute to both the underlying diseases and cirrhosis, and others may only represent susceptibility variants for underlying diseases without direct impact on cirrhosis development. A standard GWAS, which compares allele frequency differences between cases and controls, may not distinguish among these scenarios, complicating the interpretation of downstream analyses such as PheWAS, MR, and gene prioritization. For instance, PNPLA, TM6SF2, and MTARC1 are well-known NAFLD loci. Is it possible to discern if these loci showed association merely because NAFLD is common among cirrhosis patients, or do these loci directly influence cirrhosis development? If they are merely NAFLD variants, the value of downstream analysis such as gene prioritization in cirrhosis research is limited. Also, does the HLA locus only identified in cross-ancestry GWAS indicate viral hepatitis locus, or does it truly affect developing cirrhosis?

Response: This is an interesting point that we thank the Reviewer for highlighting. We agree that the different association patterns of the cirrhosis variants are relevant to address in more detail. Two new analyses are included in the revised paper:

First, we plotted the relationship between effects on cirrhosis and NAFLD of the 36 cirrhosis risk variants and 18 previously reported NAFLD variants (new **Figure 3**). The intent with these plots is to depict the heterogeneous associations in a simple, easy-to-understand way. The plots reveal several known patterns. For example, variants in *PNPLA3* and *MTARC1* affect the full NAFLD range, variants in *GCKR* associated only with NAFLD, whereas *HSD17B13* predominantly affects the later, progressive stages. The plot also clarifies that many of the variants influence cirrhosis via pathways other than fatty liver disease, e.g. variants in *SERPINA1*, *HFE*, *HLA*, and newly identified variants in/near *HKDC1*, *GYPC*, *PDE4B*, *MAMSTR*, and *HAAO*.

Second, we have now included multivariable Mendelian randomization (MVMR) analyses, to dissect out the causal independent effects of BMI and alcohol on cirrhosis. In this analysis, we conditioned the genetic effects of BMI and alcohol on cirrhosis by their genetic effect on NAFLD. We found evidence to support a causal role of BMI and alcohol on cirrhosis, which is in line with previous observational data.

In a broader context, the challenges related to dissecting out the effects of underlying risk factors from those acting directly on cirrhosis are not unique to this disease. Other complex diseases are also influenced by a spectrum of risk factors (each with its own underlying genetic determinants) as well as variants that act directly on the disease. For example, risk of coronary artery disease (CAD) is causally affected by lipid levels, obesity, smoking, and diabetes, and genetic variants that affect these risk factors therefore also affect CAD. There are also numerous genetic variants that act on CAD without affecting known risk factors (e.g. the well-known chromosome 9 locus). In our view, cirrhosis is an excellent example of how the genetic risk variants for an end-stage, complex disease act via environmental risk factors (alcohol, obesity), earlier stages of the disease (NAFLD, NASH), and likely also directly via the pathogenic driver of the disease (fibrosis). The identification of variants that influence progression to cirrhosis *per se* are of particular interest, given the lack of drugs that can either halt or even reverse the disease.

As with the two variants associated with cirrhosis mediated by their association with lower alcohol intake (in *ADH1B* and *ALDH2*), the HLA-association with cirrhosis is likely mediated via increased persistence and chronicity of viral hepatitis in carriers of this variant. This has been discussed on Page 12, Line 20, and Page 13, Line 19. In support of this notion, *HLA* is associated with chronic hepatitis B (beta=-0.33, P=2.5E-27) and cirrhosis (beta=-0.18, P=5.7E-10), but not with other risk factors for cirrhosis, including alcohol intake or obesity in Biobank Japan (<https://pheweb.jp/variant/6-33024654-A-C>).

Please see the following changes:

Methods, Page 17, Line 19: “Finally, we explored whether the genetic effects of BMI and alcohol were mediated by the effect of NAFLD, by using the `ivw_mvmmr()` function in the MVMR package. Genetic effects on NAFLD were obtained from a meta-analysis comprising 9,491 cases.¹¹”

Results, Page 7, Line 10: “**Comparison between genetic effects on NAFLD, ALD and cirrhosis.** Next, we compared the effect sizes on cirrhosis and NAFLD ($n_{\text{cases}} = 22,944$) of 18 previously reported NAFLD variants along with the 36 cirrhosis variants identified here (totaling 38 distinct variants). Eight variants had significantly higher effects on cirrhosis compared with NAFLD ($P_{\text{Het}} < 0.05/38$, Figure 3a and Supplementary Table 12). Of those, we found that rs72613567 in *HSD17B13* and known risk variants in *SERPINA1* (p.Glu366Lys) and *HFE* (p.Cys282Tyr) exhibited stronger effects on cirrhosis than on NAFLD. Moreover, we found that the newly identified variants near *HKDC1*, *HLA-DQB1* and *MAMSTR* likely influence cirrhosis via pathways distinct from those related to fatty liver disease. We also found that variants in *TRIB1*, *TM6SF2*, and *APOE* had stronger effects on NAFLD compared with cirrhosis, indicating that they may primarily exert their effect on cirrhosis via fatty liver disease. Variation in *GCKR* was strongly associated with NAFLD but had no effect on cirrhosis. The previously reported NAFLD variants p.Thr165Ala in *MTARC1* and p.Ile148Met in *PNPLA3* had proportional effects on cirrhosis. We then compared the effects of the 36 cirrhosis variants with their respective effects on alcoholic liver disease (ALD, $n_{\text{cases}} = 2,931$) and NAFLD ($n_{\text{cases}} = 22,944$, Figure 3b). We found proportional effects between NAFLD and ALD, except for three variants (p.His48Arg in *ADH1B*, $P_{\text{Het}} = 4.4 \times 10^{-14}$; rs28636836 in *HSD17B13*, $P_{\text{Het}} = 1.8 \times 10^{-4}$; and rs28712821 in *KLB*, $P_{\text{Het}} = 7.9 \times 10^{-4}$), which had significantly larger effects on ALD risk compared with NAFLD ($P_{\text{Het}} < 0.05/36$).”

Results, Page 7, Line 31: “To evaluate the potential mediating effect of NAFLD on the association between higher BMI, alcohol intake, and cirrhosis, we conducted multivariable Mendelian randomization (MVMR) while accounting for the influence of NAFLD. Despite observing a slight attenuation in the effect estimates, we observed significant independent associations between higher BMI ($\beta = 0.252$ s.d. units, $SE = 0.048$, $P_{\text{IVW}} = 2.7 \times 10^{-7}$) as well as alcohol intake ($\beta = 0.971$ s.d. units, $SE = 0.186$, $P_{\text{IVW}} = 1.3 \times 10^{-6}$) and risk of cirrhosis.”

Discussion, Page 12, Line 12: “The majority of variants were associated with hepatic lipid metabolism and fatty liver disease, with certain variants (in *GCKR*, *APOE*, and *TRIB1*) displaying significantly larger effects on NAFLD compared with cirrhosis. Other variants, such as those in *HSD17B13* and *MAMSTR*, were found to have larger effects on cirrhosis compared with NAFLD, indicating a **more dominant role in the progression to clinically advanced stages of chronic liver disease**. Variants near *HKDC1* were mainly associated with cirrhosis, indicating

the involvement of potential profibrotic pathways that does not involve the accumulation of hepatic fat. Conversely, other variants in *COBLL1* and *SH2B3* were related to body fat distribution traits, indicating that an impaired ability to store adipose tissue in peripheral compartments may contribute to disease.¹⁷⁻¹⁹ Additionally, lead variants at *ADH1B* and *ALDH2* have been shown to cause adverse symptoms with alcohol intake, thus reducing the risk of alcohol-related diseases such as cirrhosis.²⁰ The variants at *HFE* and *SERPINA1* cause hemochromatosis and alpha-1 antitrypsin deficiency, respectively, both well-known risk factors for cirrhosis. Overall, these genetic associations highlight the significant role of fat, alcohol, and iron in the pathogenesis of cirrhosis.”

Figure 3. Comparison between non-alcoholic fatty liver disease (NAFLD), alcoholic liver disease (ALD) and cirrhosis. **a**, Shown are the effects of 18 previously reported NAFLD variants and 36 cirrhosis variants identified in this study, totaling 38 distinct signals. The NAFLD effects were derived from meta-analysis of data from deCODE, UKB, CHB, Intermountain Healthcare and FinnGen (n=22,944). Cirrhosis effects were derived from the cross-ancestry meta-analysis. Variants that had stronger effects ($P_{\text{Het}} < 0.05/38$) on NAFLD compared with cirrhosis are colored blue, while variants with stronger effects on cirrhosis compared with NAFLD are colored red. **b**, Shown are the effects of the 36 cirrhosis variants on NAFLD (n=22,944) and ALD (n=2,931). Variants with stronger effects ($P_{\text{Het}} < 0.05/36$) on ALD compared with NAFLD are colored red. For both **a**, and **b**, the solid line represents the line of best fit. The dashed

identity line ($Y=X$) is shown for reference. Error bars correspond to 95% Cis. P_{Het} , P-value for heterogeneity; UKB, UK Biobank; CHB, Copenhagen Hospital Biobank; NAFLD, non-alcoholic fatty liver disease; ALD, alcoholic liver disease.

2. The authors identified 15 significant cirrhosis signals and included an additional 21 signals, selected from SNPs significantly associated with ALT or GGT. Their p-values for cirrhosis showed a false discovery rate (FDR) of less than 0.05. However, this approach appears operationally influenced. I believe such results would benefit from replication for the target trait, following the approach used in the Emdin et al. paper. Also I think it might be more natural to conduct this analysis using MTAG.

Response: We agree that independent replication would strengthen the study, as also pointed out by Reviewer #1 (comment #1). We have therefore performed a validation analysis in the Million Veteran Program (MVP) cohort which includes 21,689 cirrhosis cases and 617,729 controls. Of the 36 variants associated with cirrhosis in our original study, 20 were associated with cirrhosis at $P < 0.05$ and consistent direction of effect in the MVP cohort. Of the 15 variants that did not replicate at $P < 0.05$, we found a high level of concordance in the directions of effects between the two datasets.

Please note that the Asian-specific risk variant in *ALDH2* was not amenable to replication owing to the low number of participants of East Asian ancestry in the MVP cohort. However, the association of this variant with reduced alcohol intake in Asian populations has been firmly established since the 1980's. The variant has also been associated with protection from alcohol-related liver disease in numerous studies (e.g. PMIDs: 7904979, 29779728, 11748356, 37378938, 27189280). These associations mitigate the need for replication of its association with cirrhosis. Moreover, the *HLA* variant was unlikely to replicate in MVP given that its association with cirrhosis in our GWAS was entirely driven by Biobank Japan. The association of this variant with cirrhosis is likely secondary to its association with risk of chronic hepatitis in Asian populations.

The validation data have been incorporated into the revised manuscript as follows:

Abstract, Page 3, Line 2: 'Using data from 12 cohorts, including 18,265 cases with cirrhosis, 1,782,047 controls, up to 1 million individuals with liver function tests, and a validation cohort of 21,689 cases and 617,729 controls, we identify and validate 20 risk associations for liver cirrhosis.'

Methods, Page 16, Line 1: "**Validation.** To validate our findings, we performed replication of cirrhosis variants identified via cirrhosis GWAS and/or endophenotype-informed analysis using summary statistics of 21,689 cirrhosis cases and 617,729 disease-free controls from the MVP. Two variants (rs146650659 and rs113469203) were not available in the MVP, for which we selected suitable proxies ($r^2 \geq 0.65$). rs671 in *ALDH2* was not amenable to validation, due to low frequency in non-East Asian populations. Cases were defined as in the primary GWAS analysis (Supplementary Table 1). A $P < 0.05$ and consistent direction of effect was considered successful replication."

Results, Page 6, Line 11: "**Validation.** A total of 36 risk variants were identified through cirrhosis GWAS and/or the endophenotype-informed analysis, of which 35 were available for replication in the Million Veterans Program (MVP) cohort (21,689 cases and 617,729 controls). Replication of *ALDH2* p.Glu504Lys (rs671) was not possible due to absence in non-East Asian populations. Twenty

variants (57 %) showed significant associations ($P < 0.05$) with cirrhosis and concordant directions of effect in MVP (Supplementary Table 10). Ten of the replicated variants were initially identified in stage 1 or 2 of the GWAS, whereas 10 were identified solely via endophenotype-informed analyses. Of the 15 variants that did not replicate at $P < 0.05$, we found a high level of concordance in the magnitudes and directions of effects between the two datasets (Pearson's $r^2 = 0.73$, $P = 1.8 \times 10^{-3}$; Supplementary Table 10)."

Discussion, Page 12, Line 3: "Our study included over 18,000 cirrhosis cases and more than one million individuals with endophenotypic data sampled from four populations and identified 36 risk variants for cirrhosis of which 20 replicated in an independent cohort."

The validation results are shown in a new **Supplementary Table 10**.

Regarding MTAG: Agreeing with the Reviewer, utilizing the MTAG approach, as demonstrated in the Emdin *et al.* paper (PMID 33310085), could enhance statistical power for discovery. However, following the guidelines established by the MTAG paper (PMID 29292387), the developers recommend employing MTAG analyses between trait pairs with a strong genetic correlation ($r^2 > 0.7$). In our case, the r^2 values between cirrhosis and ALT were 0.39 (SE 0.05, $P = 1.1 \times 10^{-13}$) and 0.44 (SE 0.06, $P = 8.0 \times 10^{-14}$) with GGT.

As stated in the MTAG paper, conducting analyses with trait-pairs with lower genetic correlations comes at a risk of inflation and potential false positives. To investigate this, we ran MTAG analyses using default settings (<https://github.com/JonJala/mtag/wiki/Tutorial-1:-The-Basics>) using European ancestry cirrhosis meta-analysis ($N_{\text{case}} 15,225$) and ALT ($N = 1,010,710$) summary data. We identified 97 independent risk variants, with notably 43 variants showing $P_{\text{cirrhosis_EUR}} > 0.05$ in the primary GWAS analysis but $P_{\text{ALT}} < 0.05$ in the ALT GWAS meta-analysis. Only three variants exhibited a smaller P_{MTAG} than both $P_{\text{cirrhosis}}$ and P_{ALT} . Among the most significant loci identified by MTAG was rs35968570 ($P_{\text{MTAG}} 2.1 \times 10^{-50}$), located upstream of the *GPT* gene, which encodes alanine transferase, ie. plasma ALT, a noncausal biomarker for liver disease. Of note, only 4 out of 8 variants identified through MTAG in the Emdin *et al.* paper were validated in the present study. These findings highlight that, in situations where trait correlations fall below the recommended threshold, associations identified using MTAG are at risk of being mainly driven by the larger power in the quantitative trait.

3. The interpretation of MR result is further challenging. As many cirrhosis loci are shared with NAFLD, do the causal roles of BMI and alcohol intake for cirrhosis or for NAFLD? Can the authors clarify them?

Response: We agree with the Reviewer that variants that are shared between NAFLD and cirrhosis may complicate the interpretation. To explore whether the associations between higher BMI and alcohol intake with cirrhosis is mediated through their effects on NAFLD, we conducted multivariable Mendelian randomization analyses (MVMR) to account for NAFLD effects. In this analysis, we employed previously published NAFLD summary statistics (PMID 36280732). **While we observed a slight attenuation in the effect estimates, the associations remained statistically significant ($P_{BMI} = 2.7 \times 10^{-7}$ and $P_{alcohol} = 1.3 \times 10^{-6}$).**

Methods, Page 17, Line 19: “Finally, we explored whether the genetic effects of BMI and alcohol were mediated by the effect of NAFLD, by using the `ivw_mvmmr()` function in the MVMR package. Genetic effects on NAFLD were obtained from a meta-analysis comprising 9,491 cases.¹¹”

Results, Page 7, Line 31: “To evaluate the potential mediating effect of NAFLD on the association between higher BMI, alcohol intake, and cirrhosis, we conducted multivariable Mendelian randomization (MVMR) analyses while accounting for the influence of NAFLD. Despite observing a slight attenuation in the effect estimates, we still identified significant independent associations between higher BMI ($\beta = 0.252$ s.d., SE = 0.048, $P_{IVW} = 2.7 \times 10^{-7}$) as well as alcohol intake ($\beta = 0.971$ s.d., SE = 0.186, $P_{IVW} = 1.3 \times 10^{-6}$) and risk of cirrhosis.”

4. The interpretation of the MR results presents additional challenges. Given that many cirrhosis loci are shared with NAFLD, it becomes difficult to discern whether the causal roles of BMI and alcohol intake are directly causal for cirrhosis or for NAFLD. I would appreciate if the authors could provide some discussion.

Response: This comment is related to comment #3. Please see response above.

5. Regarding the gene burden test conducted on 19 genes, I propose a subtitle change for this section, as the standard goal of a rare variant burden test might be to identify genes not detected by the common variant association test. These 19 genes were already identified. By the way, it would be helpful if the authors could clarify their selection criteria. The analysis provides an interesting interpretation of the protective or risk effects of each gene variant. If this is the focus, it might be more beneficial to evaluate not only these 19 genes but also other genes within the common variant association loci that are in LD with GWAS lead variants, for a fine mapping purpose.

Response: We concur that it should be clarified that the 19 genes were already identified. Accordingly, we have renamed the subtitle in the Results and Methods section to ‘**Convergence**

between common and rare variant associations'. Results, Page 9, Line 9 and Methods, Page 19, Line 3.

6a. Even with the complications posed by cirrhosis GWAS as mentioned above, the construction of a PRS is meaningful if validated by an independent sample set. The PRS in this study was constructed without the UK dataset and later evaluated in the UKB dataset, thus making the evaluation scheme valid. The performance of the cirrhosis-PRS was superior when compared to the ALT-PRS. Was the improved performance in the AUC statistically significant?

Response: We agree with the Reviewer that given that the confidence intervals overlap, a formal test is needed to highlight whether the AUCs are significantly different from one another. In the revised paper, a PRS based on 15 SNPs (PRS_{15-SNP}) showed a slightly higher AUC compared to the genome-wide cirrhosis PRS. We therefore use PRS_{15-SNP} as the main PRS in the revised paper. We compared the AUC of a model with PRS_{15-SNP} (i.e. PRS_{15-SNP} + age + sex + 10 PCs) with that of a model with PRS_{ALT} (i.e. PRS_{ALT} + age + sex + 10 PCs), using the roc_test() function in the pROC R-package. We computed 95% CIs using a stratified bootstrap with 1,000 replicates. The PRS_{15-SNP} had a statistically significantly higher AUC than the PRS_{ALT}.

Results, Page 10, Line 8: “We found that the PRS_{15-SNP} explained the highest proportion of phenotypic variation ($r^2 = 1.7\%$; **Supplementary Table 9**), change in AUC (+0.031, 95% CI 0.023-0.039; **Supplementary Table 24**) and yielded an OR for cirrhosis of 1.42 per s.d. increase in PRS (**Supplementary Table 9** and **Figure 5A**). In comparison, the PRS_{ALT} accounted for 1.3% of cirrhosis phenotypic variance, had a change in AUC of 0.021 and an OR of 1.38 per s.d. increase in PRS (**Figure 5**). The difference in predictive ability between PRS_{15-SNP} and PRS_{ALT} was statistically significant (change in AUC +0.005, 95% CI 0.003-0.017, $P = 0.005$).”

6b. The authors could also evaluate the NRI.

Response: NRI analyses have been added to the revised paper.

Results, Page 10, Line 13: “Next, we evaluated reclassification of individuals after addition of the PRS_{15-SNP} to a baseline model containing age, sex, and 10 PCs. Adding PRS_{15-SNP} resulted in a net percentage of individuals with cirrhosis correctly classified upward (event net reclassification index [NRI]) of 8.4% (95% CI 3.1–13.7), and of individuals without cirrhosis correctly classified downward (non-event NRI) of 21.3% (95% CI 20.1–22.7). These changes resulted in an overall continuous NRI of 29.7% (95% CI 23.4–36.1).”

6c. Lastly, the reported higher risk of developing HCC among cirrhosis patients and developing cirrhosis among NAFLD patients remain significant findings.

Response: Thank you for the comment. To strengthen these progression analyses, we now repeat them in two independent cohorts, with similar results.

Methods, Page 20, Line 17: “The PRS was evaluated in the UKB and validated in both CHB (NAFLD to cirrhosis and cirrhosis to HCC) and MVP (NAFLD to cirrhosis). To avoid overfitting, effects were derived from meta-analysis that did not include the test dataset.”

Results, Page 11, Line 4: “A similar pattern was observed in CHB, involving 3,253 individuals with cirrhosis, of whom 172 developed HCC. Individuals in the top 20% of the PRS had an 11% (95% CI 8.5-14.0) risk of developing HCC, compared to 5.3% (95% CI 4.4-6.3, P for difference < 0.001) for those in the bottom 80% (**Figure 6b**). Correspondingly, the PRS associated with increased risk of progressing to cirrhosis in individuals with registry-defined NAFLD (**Figure 6**). We identified 4,449 individuals in the UKB with registry-defined NAFLD, of whom 193 progressed to cirrhosis during follow-up. Individuals with a PRS_{15-SNP} in the top 20% had a 10-year risk of 11.0% (95% CI 7.1-16.0), whereas individuals in the bottom 80% of the distribution had a 10-year risk of 8.6% (95% CI 6.8-11.0, P for difference = 0.036; **Figure 6c**). In CHB, among 860 individuals with registry-defined NAFLD, 95 developed cirrhosis during follow-up. In MVP, out of the 18,302 individuals with NAFLD, 280 developed cirrhosis. Those in the top 20% of the PRS_{15-SNP} distribution had a 10-year cirrhosis risk of 13.0% (95% CI 7.5-19.0) in CHB (**Figure 6d**), and a 5-year risk of 2.8% (95% CI 2.3-3.5) in MVP (**Figure 6e**), respectively. In contrast, those in the bottom 80% of PRS_{15-SNP} had a 10-year risk of 9.9% (95% CI 7.6-12.0, P for difference = 0.032) in CHB and a 5-year risk of 1.5% (95% CI 1.3-1.7, P for difference < 0.001) in MVP, respectively.

Minor points:

Several technical comments on the MR analysis:

7. To minimize weak instrumental bias, the authors excluded traits with fewer than ten instrumental variables. I recommend performing an F-test to further mitigate this issue.

Response: F-statistics have now been included in **Supplementary Table 13**.

Methods, Page 17, Line 9: “We evaluated instrument strength by calculating the F-statistic.³⁶”

8. One potential violation of MR assumptions is linkage disequilibrium (LD). The authors considered r^2 in clumping to resolve this issue but did not specify the source of their r^2 data. This should be clarified.

Response: We have provided details on the reference panel used for clumping.

Methods, Page 17, Line 11: "We selected independent variants with genome-wide significance ($P < 5 \times 10^{-8}$) and an $r^2 < 0.001$ to serve as instrumental variables (IVs) for our MR analyses using the clumping procedure in the TwoSampleMR software and European 1000 Genomes reference panel."

9. There might be a potential bias introduced by using outcome summary statistics with LD structures slightly different from European ancestry such as Biobank Japan, All of Us (Hispanic), and All of Us (African American). I encourage a discussion on possible biases from using these datasets, or the addition of some sensitivity analyses. Please consider using the same single-ancestry derived GWAS for exposure and outcome.

Response: We apologize if this was not clear in the original version of the manuscript. We did indeed only include European ancestry samples in both exposures and outcome data to mitigate any bias attributed to differences in LD structure. We have added the following:

Methods, Page 17, Line 5: "To ensure that our analysis did not have overlapping samples, we conducted a meta-analysis on all available cirrhosis cohorts of European ancestry except for the UKB sample set, as all the exposure traits were derived from the UKB. We excluded exposure traits with fewer than ten instrumental variables to avoid underpowered tests, resulting in 39 traits being tested. We evaluated instrument strength by calculating the F-statistic.³⁶ To ensure comparable LD structure between exposure and outcome datasets, only exposures derived from samples of European ancestry were taken forward."

10. The data sources for exposure body mass index (BMI) and alcohol were not provided. Please clarify these and their source populations.

Response: We have now provided references and source population in **Supplementary Table 13**.

11. In the methods section, the authors mentioned, “We used two different MR methods: inverse variance weighted (IVW) model as our primary model and the weighted median model as sensitivity analysis. MR-Egger-intercept was used to for pleiotropy.” However, I noticed results from MR-PRESSO in Supplementary Figures 1 and 2. This discrepancy needs to be resolved.

Response: We apologize for this. The MR-PRESSO regression line is a standard output from the `mr_scatter_plot()` function in the TwoSampleMR package. We have now revised these plots and only included regression lines from the IVW, the weighted median model and MR Egger.

13. For SNP heritability analysis, a clearer description of the method would be beneficial, instead of stating "default settings".

Response: We have now provided more details on how the SNP heritability was estimated. Please see below.

Methods, Page 17, Line 23: “We used LDSC v. 1.0.0, to estimate the SNP heritability of cirrhosis in Europeans, East Asians, African Americans, and Hispanics using ancestry-matched pre-computed LD scores obtained at <https://gnomad.broadinstitute.org/downloads/>. We reformatted association statistics to LDSC format with the munge-tool, which excluded variants that did not match with the LD panel, had strand ambiguity, $MAF < 0.01$, $INFO < 0.9$, and variants that resided in long-range LD-regions and the major histocompatibility (MHC) locus on chromosome 6. To convert to liability scale, we used population-specific prevalence estimates, ranging from 0.5% in Europeans to 1.7% in East Asians.”

14. The authors found a significant gene-environment interaction of the PNPLA3 variant with alcohol intake, obesity, and T2D. The validity of testing for a gene-environment interaction term depends on the absence of a gene-environment association (Dudbridge, AJHG 2014). While I do not believe there is a strong association of this gene with these environmental factors, I recommend stating this more clearly in the text.

Response: We investigated the association between the *PNPLA3* variant (rs738408) and the environmental factors and found only weak associations with T2D (OR 1.02, $P = 0.023$) and BMI ($\beta = -0.04 \text{ kg/m}^2$, $P = 0.001$) and no association with weekly alcohol intake ($\beta = -0.04 \text{ units/week}$, $P = 0.164$). These null or weak associations agree with the literature.

Results, Page 8, Line 6: “To determine whether similar interactions exist between environmental factors and newly identified risk variants, we examined the effects of 35 genetic variants (excluding *HLA*) on cirrhosis risk in combination with environmental factors in the UKB. We found that rs738408

in *PNPLA3* interacted significantly with T2D ($P = 7.9 \times 10^{-6}$), BMI ($P = 3.0 \times 10^{-6}$), and weekly alcohol intake ($P = 1.2 \times 10^{-5}$) on the risk of cirrhosis (**Supplementary Table 14**). We found that rs738408 was only weakly associated with T2D (OR 1.03, $P = 0.007$), BMI ($\beta = -0.04 \text{ kg/m}^2$, $P = 0.001$) and not associated with weekly alcohol intake ($\beta = -0.04 \text{ units/week}$, $P = 0.160$)."

Manuscript ID: NG-A62855.R1

Manuscript title: Integrative common and rare variant analyses provide insights into the genetic architecture of liver cirrhosis

Point-by-point responses to comments from Reviewer #3.

Response: Thank you for the kind words and for providing constructive comments. Our responses are provided in blue below. Text incorporated into the revised paper is shown in red.

Remarks to the Author:

The authors perform the to date largest GWAS for the liver cirrhosis phenotype harnessing multiple international cohorts. The study is timely and well performed and will be met with substantial interest in the liver community.

I have only a couple stylistic / minor requests:

1. A traditional table with the identified loci, combined p-value and OR would be nice as part of the main manuscript.

Response: As suggested, a new **Table 1** with the above-mentioned information has now been included in the main manuscript.

2. A stronger focus on differentiation of alcoholic and non-alcoholic liver disease (are they really different?) would be helpful.

Response: This is an interesting point that was also brought up by Reviewer #2 (comment #1). We agree that a focus on the different association patterns with alcoholic and non-alcoholic liver disease of the cirrhosis variants are worth exploring.

To shed light on this, we have now provided a scatterplot showing the effects of the 36 cirrhosis variants on NAFLD versus their effects on alcoholic liver disease (ALD). We found three variants, p.His48Arg in *ADH1B*, rs28636836 in *HSD17B13*, and rs28712821 in *KLB*, that showed significantly larger effects on ALD compared with NAFLD. Please see the following changes:

Results, Page 7, Line 10: “Comparison of genetic effects on NAFLD, ALD and cirrhosis. Next, we compared the effect sizes on cirrhosis and NAFLD ($n_{\text{cases}} = 22,944$) of 18 previously reported NAFLD variants along with the 36 cirrhosis variants identified here (totaling 38 distinct variants). Eight variants had significantly higher effects on cirrhosis compared with NAFLD ($P_{\text{Het}} < 0.05/38$, Figure 3a and Supplementary Table 12). Of those, we found that rs72613567 in *HSD17B13* and known risk variants in *SERPINA1* (p.Glu366Lys) and *HFE* (p.Cys282Tyr) exhibited stronger effects on cirrhosis than on NAFLD. Moreover, we found that the newly identified variants near *HKDC1*, *HLA-DQB1* and *MAMSTR* likely influence cirrhosis via pathways distinct from those related to fatty liver disease. We also found that variants in *TRIB1*, *TM6SF2*, and *APOE* had stronger effects on NAFLD compared with cirrhosis, indicating that they may primarily exert their effect on cirrhosis via fatty liver disease. Variation in *GCKR* was strongly associated with NAFLD but had no effect on cirrhosis. The previously reported NAFLD variants p.Thr165Ala in *MTARC1* and p.Ile148Met in *PNPLA3* had proportional effects on cirrhosis. We then compared the effects of the 36 cirrhosis variants with their respective effects on alcoholic liver disease (ALD, $n_{\text{cases}} = 2,931$) and NAFLD ($n_{\text{cases}} = 22,944$, Figure 3b). We found proportional effects between NAFLD and ALD, except for three variants (p.His48Arg in *ADH1B*, $P_{\text{Het}} = 4.4 \times 10^{-14}$; rs28636836 in *HSD17B13*, $P_{\text{Het}} = 1.8 \times 10^{-4}$; and rs28712821 in *KLB*, $P_{\text{Het}} = 7.9 \times 10^{-4}$), which had significantly larger effects on ALD risk compared with NAFLD ($P_{\text{Het}} < 0.05/36$).”

Figure 3. Comparison between non-alcoholic fatty liver disease (NAFLD), alcoholic liver disease (ALD) and cirrhosis. **a**, Shown are the effects of 18 previously reported NAFLD variants and 36 cirrhosis variants identified in this study, totaling 38 distinct signals. The NAFLD effects were derived from meta-analysis of data from deCODE, UKB, CHB, Intermountain Healthcare and FinnGen (n=22,944). Cirrhosis effects were derived from the cross-ancestry meta-analysis. Variants that had stronger effects ($P_{\text{Het}} < 0.05/38$) on NAFLD compared with cirrhosis are colored blue, while variants with stronger effects on cirrhosis compared with NAFLD are colored red. **b**, Shown are the effects of the 36 cirrhosis variants on NAFLD (n=22,944) and ALD (n=2,931). Variants with stronger effects ($P_{\text{Het}} < 0.05/36$) on ALD compared with NAFLD are colored red. For both **a**, and **b**, the solid line represents the line of best fit. The dashed identity line (Y=X) is shown for reference. Error bars correspond to 95% Cis. P_{Het} , P-value for heterogeneity; UKB, UK Biobank; CHB, Copenhagen Hospital Biobank; NAFLD, non-alcoholic fatty liver disease; ALD, alcoholic liver disease

3. Supplementary tables should be formatted in landscape mode to fit on one page.

Response: The 24 supplementary tables are provided in an Excel file, with each table shown in separate sheets. Some of the tables are very large. It will not be possible to fit these on one page, either in landscape or portrait format.

Decision Letter, first revision:

21st Nov 2023

Dear Dr Ghouse,

Your Article, "Integrative common and rare variant analyses provide insights into the genetic architecture of liver cirrhosis" has now been seen by 3 referees. You will see from their comments below that while they find your work of interest, some important points are raised. We are interested in the possibility of publishing your study in Nature Genetics, but would like to consider your response to these concerns in the form of a revised manuscript before we make a final decision on publication.

As you will see from these comments, Reviewers #1 and #2 have minor concerns on the replication analysis. For Reviewer #1's point, you could present the replication results both before and after correcting for multiple testing, clearly noting which variants pass each threshold.

We therefore invite you to revise your manuscript taking into account all reviewer and editor comments. Please highlight all changes in the manuscript text file. At this stage we will need you to upload a copy of the manuscript in MS Word .docx or similar editable format.

*2) If you have not done so already please begin to revise your manuscript so that it conforms to our Article format instructions, available here.

*3) Include a revised version of any required Reporting Summary:
<https://www.nature.com/documents/nr-reporting-summary.pdf>

Please be aware of our guidelines on digital image standards.

[redacted]

We hope to receive your revised manuscript within four to eight weeks. If you cannot send it within this time, please let us know.

Sincerely,
Wei

Wei Li, PhD
Senior Editor
Nature Genetics
New York, NY 10004, USA
www.nature.com/ng

Reviewers' Comments:

Reviewer #1:

Remarks to the Author:

The authors have successfully responded to many of the comments. However, a major concern remains with the replication analysis that was not corrected for multiple testing. The replication p-values should be corrected for multiple testing as that is a standard requirement for a robust and

rigorous replication analysis in the field. Thus, in addition to having a concordant direction of effect, the replicated variants should pass a p-value of 0.05/35, i.e. $p < 0.00142857$. Looking into the new supplementary table 10 for variants passing this multiple testing corrected p-value in the concordant direction of effect seems to result in only 14 replicated variants. Thus, the correct replication rate appears to be only 40% instead of the reported 57%. This should be corrected.

Reviewer #2:

Remarks to the Author:

Thank you for the additional analyses. The authors have addressed all of my concerns. I have only one comment regarding Supplementary Table 10.

In the main text, it is stated that "A total of 36 risk variants were identified through cirrhosis GWAS and/or the endophenotype-informed analysis, of which 35 were available for replication in the MVP cohort (21,689 cases and 617,729 controls). Replication of ALDH2 p.Glu504Lys (rs671) was not possible due to its absence in non-East Asian populations." However, rs671 is listed in Supplementary Table 10. I understand that the high P value for this SNP in the MVP cohort is likely due to its very low allele frequency, which does not necessarily indicate a failure in replication. Thus, please consider either removing this particular SNP from the table, as mentioned in the text, or adding the EAF for each population to aid readers in interpreting this table.

Reviewer #3:

Remarks to the Author:

My points have been addressed to my satisfaction.

Author Rebuttal, first revision:

Manuscript ID: NG-A62855.R2

Manuscript title: Integrative common and rare variant analyses provide insights into the genetic architecture of liver cirrhosis

Reviewers' Comments:

Reviewer #1:

Remarks to the Author:

The authors have successfully responded to many of the comments. However, a major concern remains with the replication analysis that was not corrected for multiple testing. The replication p-values should be corrected for multiple testing as that is a standard requirement for a robust and rigorous replication analysis in the field. Thus, in addition to having a concordant direction of effect, the replicated variants should pass a p-value of 0.05/35, i.e. $p < 0.00142857$. Looking into the new supplementary table 10 for variants passing this multiple testing corrected p-value in the concordant direction of effect seems to result in only 14 replicated variants. Thus, the correct replication rate appears to be only 40% instead of the reported 57%. This should be corrected.

Reply: We once again thank the Reviewer for the careful review of our work, and for providing constructive comments. As suggested by the Reviewer, we have now changed the threshold for successful replication to $P < 1.4 \times 10^{-3}$, to account for multiple testing. Please see changes below:

Abstract, page 3, line 5: “Using data from 12 cohorts, including 18,265 cases with cirrhosis, 1,782,047 controls, up to 1 million individuals with liver function tests, and a validation cohort of 21,689 cases and 617,729 controls, we identify and validate 14 risk associations for liver cirrhosis.”

Methods, page 6, line 15: “A $P < 1.4 \times 10^{-3}$ (0.05/35 variants) and consistent direction of effect was considered successful replication.”

Results, page 16, line 6: “Of the 35 variants, 14 (40%) reached Bonferroni significance ($P < 1.4 \times 10^{-3}$ [0.05/35 variants]), and six were nominally significant ($P < 0.05$) with concordant directions of effect in MVP (Supplementary Table 10).”

Reviewer #2:

Remarks to the Author:

Thank you for the additional analyses. The authors have addressed all of my concerns. I have only one comment regarding Supplementary Table 10.

In the main text, it is stated that “A total of 36 risk variants were identified through cirrhosis GWAS and/or the endophenotype-informed analysis, of which 35 were available for replication in the MVP cohort (21,689 cases and 617,729 controls). Replication of ALDH2 p.Glu504Lys (rs671) was not possible due to its absence in non-East Asian populations.” However, rs671 is listed in Supplementary Table 10. I understand that the high P value for this SNP in the MVP cohort is likely due to its very low allele frequency, which does not necessarily indicate a failure in replication. Thus, please consider either removing this particular SNP from the table, as mentioned in the text, or adding the EAF for each population to aid readers in interpreting this table.

Reply: Thank you for taking the time to review our work. Since this variant is not taken forward in our replication, we agree with the Reviewer that it does not make sense to list it in Supplementary Table 10. The variant is now removed from Supplementary Table 10.

Reviewer #3:

Remarks to the Author:

My points have been addressed to my satisfaction.

Decision Letter, second revision:

1st Feb 2024

Dear Dr. Ghouse,

Thank you for submitting your revised manuscript "Integrative common and rare variant analyses provide insights into the genetic architecture of liver cirrhosis" (NG-A62855R1). It has now been seen by the original referees and their comments are below. The reviewers find that the paper has improved in revision, and therefore we'll be happy in principle to publish it in Nature Genetics, pending minor revisions to comply with our editorial and formatting guidelines.

Sincerely,
Wei

Wei Li, PhD
Senior Editor
Nature Genetics
New York, NY 10004, USA
www.nature.com/ng

Reviewer #1 (Remarks to the Author):

I have no further questions or comments.

Final Decision Letter:

18th Mar 2024

Dear Dr. Ghouse,

I am delighted to say that your manuscript "Integrative common and rare variant analyses provide insights into the genetic architecture of liver cirrhosis" has been accepted for publication in an upcoming issue of Nature Genetics.

Your paper will be published online after we receive your corrections and will appear in print in the next available issue. You can find out your date of online publication by contacting the Nature Press Office (press@nature.com) after sending your e-proof corrections.

Please note that *Nature Genetics* is a Transformative Journal (TJ). Authors may publish their research with us through the traditional subscription access route or make their paper immediately open access through payment of an article-processing charge (APC). Authors will not be required to make a final decision about access to their article until it has been accepted. Find out more about Transformative Journals

Authors may need to take specific actions to achieve compliance with funder and institutional open access mandates. If your research is supported by a funder that requires immediate open access (e.g. according to Plan S principles) then you should select the gold OA route, and we will direct you to the compliant route where possible. For authors selecting the subscription publication route, the journal's standard licensing terms will need to be accepted, including <https://www.nature.com/nature-portfolio/editorial-policies/self-archiving-and-license-to-publish>. Those licensing terms will supersede any other terms that the author or any third party may assert apply to any version of the manuscript.

If you have not already done so, we invite you to upload the step-by-step protocols used in this manuscript to the Protocols Exchange, part of our on-line web resource, natureprotocols.com. If you complete the upload by the time you receive your manuscript proofs, we can insert links in your article that lead directly to the protocol details. Your protocol will be made freely available upon publication of your paper. By participating in natureprotocols.com, you are enabling researchers to more readily reproduce or adapt the methodology you use. [Natureprotocols.com](http://natureprotocols.com) is fully searchable, providing your protocols and paper with increased utility and visibility. Please submit your protocol to <https://protocolexchange.researchsquare.com/>. After entering your nature.com username and password you will need to enter your manuscript number (NG-A62855R2). Further information can be found at <https://www.nature.com/nature-portfolio/editorial-policies/reporting-standards#protocols>

Sincerely,
Wei

Wei Li, PhD
Senior Editor
Nature Genetics
New York, NY 10004, USA
www.nature.com/ng